# Gene regulatory dynamics during craniofacial development in a carnivorous marsupial

**Laura E Cook[1]\*‡, Charles Y Feigin[2], John D Hills[1], Davide M Vespasiani[3], Andrew J Pask[1]†, Irene Gallego Romero[4,5,6]\*†**

[1]School of BioSciences, University of Melbourne, Royal Parade, Parkville, Australia; [2]School of Agriculture, Biomedicine and Environment, La Trobe University, Victoria, Australia; [3]Genetics and Gene Regulation, Walter and Eliza Hall Institute of Medical Research, Parkville, Australia; [4]Human Genomics and Evolution, St Vincent's Institute of Medical Research, Fitzroy, Australia; [5]School of Medicine, University of Melbourne, Royal Parade, Parkville, Australia; [6]Institute of Genomics, University of Tartu, Tartu, Estonia

**\*For correspondence:**
lecook@lbl.gov (LEC);
irene.gallego@svi.edu.au (IGR)

†These authors contributed equally to this work

**Present address:** ‡Environmental Genomics and Systems Biology Division, Lawrence Berkeley National Laboratory, Berkeley, United States

## eLife Assessment

This **important** study of regulatory elements and gene expression in the craniofacial region of the fat-tailed dunnart shows that, compared to placental mammals, marsupial craniofacial tissue develops in a precocious manner, with enhancer regulatory elements as primary driver of this difference. The **compelling** data, including a new dunnart genome assembly, provide an invaluable reference for future mammalian evolution studies, especially once additional developmental time point for the fat-tailed dunnart become available.

**Abstract** Marsupials and placental mammals exhibit significant differences in reproductive and life history strategies. Marsupials are born highly underdeveloped after an extremely short period of gestation, leading to prioritized development of structures critical for post-birth survival in the pouch. Critically, they must undergo accelerated development of the orofacial region compared to placentals. Previously, we described the accelerated development of the orofacial region in the carnivorous Australian marsupial, the fat-tailed dunnart *Sminthopsis crassicaudata*, that has one of the shortest gestations of any mammal. By combining genome comparisons of the mouse and dunnart with functional data for the enhancer-associated chromatin modifications, H3K4me3 and H3K27ac, we investigated divergence of craniofacial regulatory landscapes between these species. This is the first description of genome-wide face regulatory elements in a marsupial, with 60,626 putative enhancers and 12,295 putative promoters described. We also generated craniofacial RNA-seq data for the dunnart to investigate expression dynamics of genes near predicted active regulatory elements. While genes involved in regulating facial development were largely conserved in mouse and dunnart, the regulatory landscape varied significantly. Additionally, a subset of dunnart-specific enhancers was associated with genes highly expressed only in dunnart relating to cranial neural crest proliferation, embryonic myogenesis, and epidermis development. Comparative RNA-seq analyses of facial tissue revealed dunnart-specific expression of genes involved in the development of the mechanosensory system. Accelerated development of the dunnart sensory system likely relates to the sensory cues received by the nasal–oral region during the postnatal journey to the pouch. Together, these data suggest that accelerated face development in the dunnart may be driven by dunnart-specific enhancer activity. Our study highlights the power of marsupial–placental

comparative genomics for understanding the role of enhancers in driving temporal shifts in development.

## Introduction

The vertebrate head is a highly complex region of the body that plays a key role in an organism's ecology by centralizing numerous structures involved in diet, sensory perception, and behavior (*Francis-West and Crespo-Enriquez, 2016*). Consequently, evolution has modified craniofacial development across lineages, producing a wide array of head morphologies concomitant with the diverse niches that vertebrates occupy. Craniofacial diversity among the major mammalian lineages, in particular, has long been of great interest, due to the striking differences in their developmental ontology.

Placental mammals are characterized by a long gestation with a high maternal investment during pregnancy, and a considerable degree of both orofacial and neurocranial development occurring in the embryo in utero. As a result, placental young experience little functional constraint during early developmental stages. By contrast, marsupials have a short gestation and give birth to highly altricial young that must crawl to the teat, typically located within the maternal pouch, where they complete the remainder of their development ex utero. This unique reproductive method is thought to have imposed strong pressures on the evolution and development of the limbs and head. In particular, marsupials show accelerated development of the nasal cavity, tongue, oral bones, and musculature relative to the development of the posterior end of the body (*Nunn and Smith, 1998*; *Smith, 2006*; *Smith, 2001a*; *Cook et al., 2021*), and generally when compared to placental embryonic development (*Nunn and Smith, 1998*; *Smith, 2006*; *Smith, 1997*). Additionally, aspects of the peripheral nervous system appear to be accelerated in the face. We and others have observed large medial nasal swellings early in marsupial development that are innervated and proposed to be necessary for the sensory needs of newborn young in their journey to the pouch (*Veitch et al., 2000*; *Jones and Munger, 1985*; *Waite et al., 1994*). Comparative morphometric studies have provided a wealth of evidence that this stark difference in craniofacial development has imposed different regimes of constraint on marsupial and placental mammals (*Nunn and Smith, 1998*; *Smith, 2006*; *Cook et al., 2021*; *Smith, 1997*; *Spiekman and Werneburg, 2017*; *Koyabu et al., 2014*; *Goswami et al., 2016*) with marsupials in particular showing significantly less interspecies variation in orofacial structures and nasal morphology than placental mammals (*Goswami et al., 2016*; *Sears, 2004*; *Hüppi et al., 2018*). In spite of these observations, the molecular mechanisms that underlie these differences in early craniofacial development between marsupial and placental mammals remain poorly understood.

Cis-acting regulatory regions have been proposed to play a significant role in morphological divergence in the face, with a number of well-described enhancers that fine-tune face shape in mammals (*Attanasio et al., 2013*). There is also some evidence of a role for regulatory regions in craniofacial heterochrony in marsupials. One recent study found a marsupial-specific region within a *Sox9* enhancer that drives early and broad expression in pre-migratory neural crest cell domains contributing to early migration of cranial neural crest cells relative to the mouse (*Wakamatsu and Suzuki, 2019*; *Wakamatsu et al., 2014*). However, no study has thus far attempted to compare the overall regulatory landscape between marsupials and placentals at developmentally comparable stages. Such surveys have the potential to provide functional insights into the loci controlling craniofacial heterochrony in mammals and consequently the causative evolutionary changes in the genome that have driven the divergent ontogenies of marsupials and placentals.

In recent years, the fat-tailed dunnart (*Sminthopsis crassicaudata*, hereafter referred to as the dunnart) has emerged as a tractable marsupial model species (*Cook et al., 2021*; *Suárez et al., 2017*). Dunnarts are born after 13.5 days of gestation and craniofacial heterochrony in line with what has been reported in other marsupials is readily observable (*Nunn and Smith, 1998*; *Smith, 2006*; *Cook et al., 2021*; *Smith, 1997*; *Spiekman and Werneburg, 2017*; *Koyabu et al., 2014*; *Goswami et al., 2016*), making this species an excellent system for comparative studies with placental models. The dunnart provides an ideal model in which to study the regulatory landscape during craniofacial development in marsupials, as they rely entirely on the advanced chondrocranium with bony elements of the skeleton not present until approximately 24 hr after birth (*Cook et al., 2021*). To investigate a potential role for regulatory elements in this heterochrony, we used chromatin immunoprecipitation (ChIP)-sequencing and RNA-sequencing on craniofacial tissue (fronto-nasal, mandibular, and maxillary

**eLife digest** Marsupials are a distinctive group of mammals best known for their defining trait: a pouch. Unlike monotremes, which lay eggs, and placental mammals, in which young develop fully in the womb, marsupials give birth to highly premature offspring. These young complete most of their development within the mother's pouch.

Because of this strategy, marsupial newborns possess traits critical for immediate survival outside the womb. These include well-developed forelimbs for climbing into the pouch and a mature oral region for attaching to the teat and suckling.

Evolution has shaped the development of the head to match the diverse environments vertebrates inhabit. Marsupials diverged from placental mammals over 160 million years ago, and their requirement for early limb and oral development makes them a powerful system for investigating the genetic mechanisms underlying development.

The fat-tailed dunnart (Sminthopsis crassicaudata), often called a marsupial mouse, is emerging as an important model organism. With its short gestation period and extremely undeveloped state at birth, the dunnart provides an excellent comparison to the laboratory mouse – a well-established placental model – for studying evolutionary and developmental differences.

Cook et al. investigated the genes and regulatory elements driving early orofacial development in the dunnart, comparing their findings with existing craniofacial expression and epigenomic data in mice.

Their results showed that although the genes involved in craniofacial development are highly conserved between the two species, the regulatory elements controlling those genes differ markedly. In dunnarts, regulatory elements linked to skin, nervous system, and muscle development were highly active, whereas in mice they were inactive or only weakly expressed.

An important feature of marsupial development is the postnatal retention of the periderm (a transient outer cell layer in embryonic skin). This may support gas exchange in prematurely born young. In addition, elevated activity of genes regulating sensory and muscle development suggests early maturation of mechanosensory and olfactory systems, which are essential for the newborn's journey to the pouch. Notably, pharyngeal and facial muscles develop before the skeletal system, contrasting with the developmental sequence in placental mammals.

In summary, marsupials exhibit a unique reproductive strategy, in which young are born in an extremely underdeveloped state and survival depends on accelerated development of specific traits after birth. By comparing gene regulation in marsupials and placentals, researchers gain insights into how evolution shaped developmental pathways. Expanding genomic resources for the dunnart, such as genome editing tools and transgenic models, will further enhance its role as a powerful comparative system for evolutionary developmental biology.

prominences) collected from newborn dunnart pouch young. We performed a detailed characterization of chromatin marks during early craniofacial development and then comparative analyses with the placental laboratory mouse. Our work provides valuable insights into genomic regions associated with regulatory elements regulating craniofacial development in marsupials and their potential role in craniofacial heterochrony.

## Results

### Defining craniofacial putative enhancer and promoter regions in the dunnart

After validating the ability of our antibodies to enrich for dunnart chromatin marks (see Appendix), ChIP-seq libraries were sequenced to average depth of 57 million reads and mapped to a de novo assembly of the dunnart genome generated for this study (see Methods). Peak calling with MACS2 ($q < 0.05$) identified 80,989 regions reproducibly enriched for H3K4me3 and 121,281 regions reproducibly enriched for H3K27ac in dunnart facial prominence tissue. As this is the first epigenomic profiling for this species, we performed extensive data quality control to ensure the robustness of the data. Similar to previous studies (*Villar et al., 2015*; *Heintzman et al., 2009*; *Zhu et al., 2013*; *Cain et al.,*

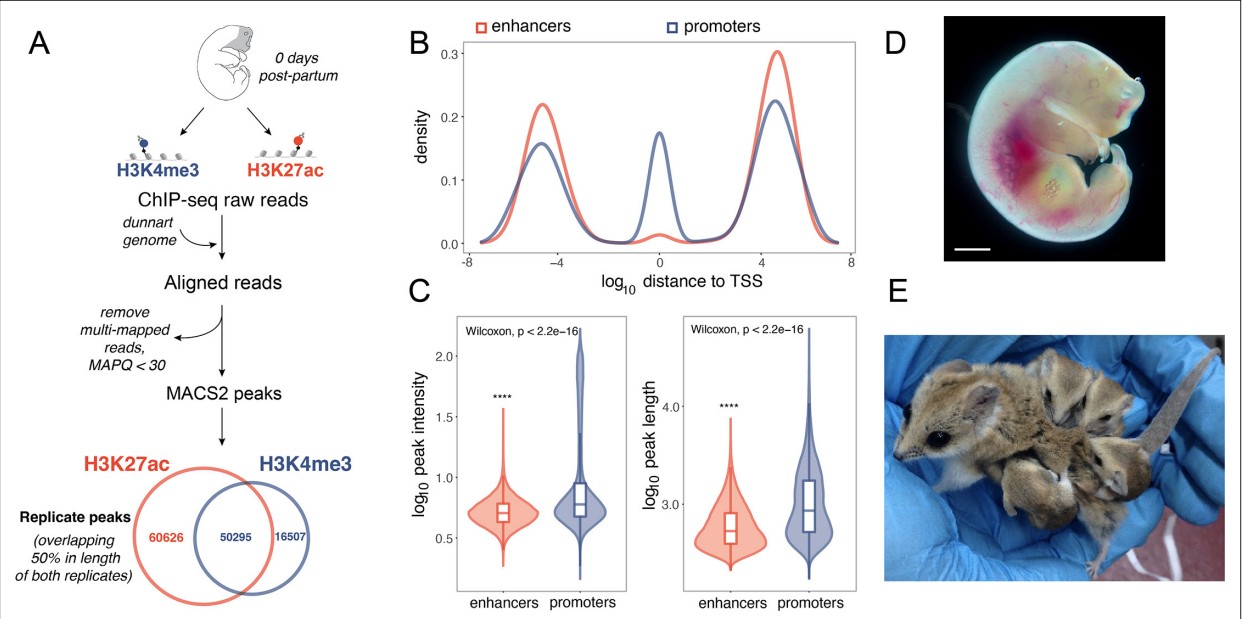

**Figure 1.** Analysis workflow and quality control of H3K4me3 and H3K27ac peaks in the fat-tailed dunnart (*Sminthopsis crassicaudata*). (**A**) Drawing of a D0 dunnart pouch young with dissected orofacial tissue shown in gray. Short-read alignment and peak calling workflow and numbers of reproducible peaks identified for H3K27ac (orange), H3K4me3 (blue) for craniofacial tissue. (**B**) $\log_{10}$ distance to the nearest TSS for putative enhancer (orange) and promoters (blue). (**C**) $\log_{10}$ of peak intensity and peak length are represented as boxplots and violin plots for putative enhancers (orange) and promoters (blue). Peak intensities correspond to average fold enrichment values over total input DNA across biological replicates. (**D**) Dunnart pouch young on the day of birth. Scale bar = 1 mm. (**E**) Adult female dunnart carrying four young. Statistical significance (Wilcoxon, FDR-adjusted, $p < 0.00001$) compared to cluster 1 promoter-associated peaks is denoted by ****.

*2011*; *Santos-Rosa et al., 2002*), we found that H3K4me3 often (62% of all H3K4me3 peaks) co-occupied the genome with H3K27ac (*Figure 1A*) while 50% of H3K27ac-enriched regions were only associated with this mark (*Figure 1A*). Active enhancers are generally enriched for H3K27ac (*Villar et al., 2015*; *Creyghton et al., 2010*) while sites of transcription initiation (active promoters) can be identified as being marked by both H3K27ac and H3K4me3 (*Cain et al., 2011*; *Santos-Rosa et al., 2002*).

We initially defined promoters as those marked by only H3K4me3 or with >50% of reciprocal peak length for H3K27ac and H3K4me3 peaks, and enhancers as those marked only by H3K27ac, identifying 66,802 promoters and 60,626 enhancers. Enhancers were located on average 77 kb from TSS, while promoters were located on average 106 kb from the nearest TSS, despite there being a greater number of peaks located <1 kb from the TSS (1008 enhancers vs. 9023 promoters; *Figure 1B*). This was an unexpected finding as a large fraction (0.41) of promoters were located >3 kb from an annotated TSS (see Appendix). H3K4me3 activity at enhancers is well established (*Koch and Andrau, 2011*; *Pekowska et al., 2011*); however, compared to H3K4me3 activity at promoters, H3K4me3 levels at enhancers are low (*Calo and Wysocka, 2013*). This is in line with our observations that H3K4me3 levels at enhancers were nearly seven times lower than those observed at promoter regions (see Appendix for details). Distance from TSS is frequently used to filter putative promoters from other elements, therefore, we grouped peaks characterized as promoters based on their distance to the nearest TSS (see Methods), resulting in 12,295 high-confidence promoters for all of the following analyses (see Appendix).

## Candidate dunnart regulatory elements are associated with highly expressed genes involved in muscle, skin, and bone development

Next, we asked what biological processes are associated with active regulatory regions in the dunnart face. To accomplish this, we linked peaks to genes in order to associate functional annotations of coding genes with the candidate regulatory elements that likely regulate their expression. To make use of resources available in model organisms such as GO databases, we converted all dunnart gene IDs to mouse orthologous genes for downstream applications. This reduced the dataset to 35,677

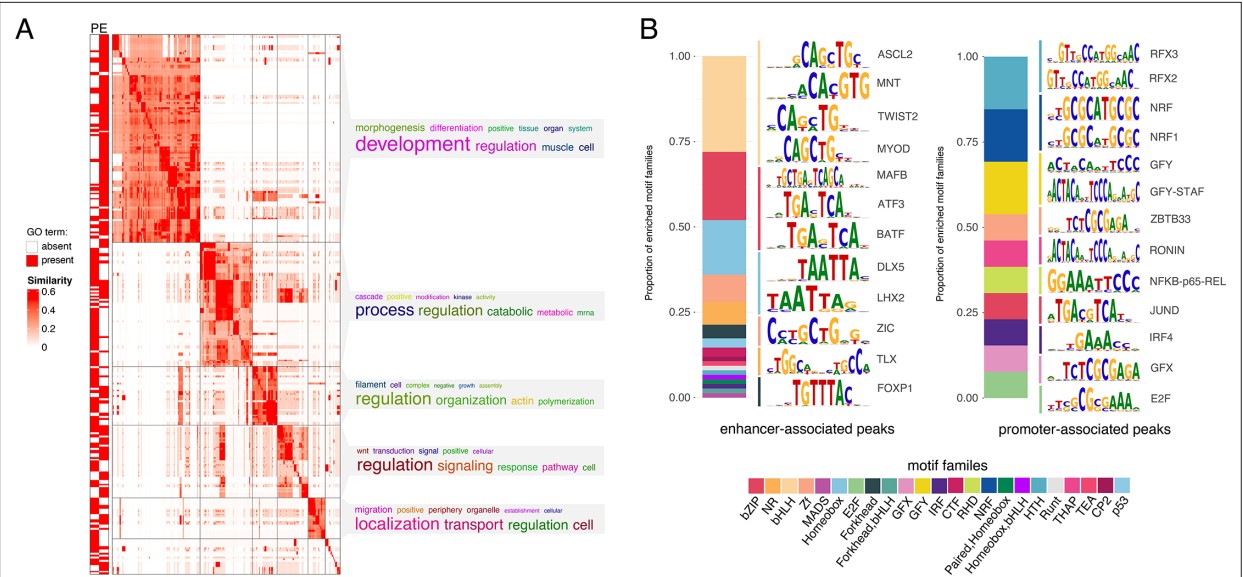

**Figure 2.** Predicted functional enrichment for dunnart peaks. (**A**) 304 significantly enriched GO terms clustered based on similarity of the terms. The function of the terms in each group is summarized by word clouds of the keywords. Rows marked by P were driven by genes linked to putative promoters, rows marked by E were driven by genes linked to putative enhancers. (**B**) Enriched TF motifs for transcription factor families (HOMER). PWM logos for preferred binding motifs of TFs are shown. The letter size indicates the probability of a TF binding given the nucleotide composition.

The online version of this article includes the following figure supplement(s) for figure 2:

**Figure supplement 1.** Homer motif enrichment for dunnart promoters and enhancers for the top 20 enriched TF families.

putative enhancers and 8589 promoters near (within 1 Mbp) genes with a one-to-one ortholog in mouse (***Supplementary file 1a***).

We found that gene annotations for both promoters and enhancers were enriched for 23% of the same GO terms, including cellular processes (protein localization to plasma membrane, protein localization to cell periphery, regulation of cell morphogenesis, positive regulation of cell migration) and development (axon development, camera-type eye development, muscle tissue development, striated muscle development). By contrast, 44% of GO terms were uniquely enriched among genes assigned to promoters and were related to mRNA processing, transcription, mRNA stability, cell cycle, and mRNA degradation (***Figure 2A***, ***Supplementary file 1b***) and uniquely enriched GO terms for genes assigned to candidate enhancers corresponded to processes indicative of early embryonic development (***Figure 2A***, ***Supplementary file 1c***).

Terms related to facial skeleton development were enriched among genes assigned to putative dunnart enhancers, including bone cell development, muscle cell development, secondary palate development, roof of mouth development, and mesenchyme development, consistent with dunnart craniofacial morphology (***Cook et al., 2021***). Enhancers active near important palate genes (***Won et al., 2023***; ***Zarate et al., 2024***) such as *SHH*, *SATB2*, *MEF2C*, *SNAI2*, and *IRF6* in the dunnart at birth may highlight potential regulatory mechanisms driving early palatal closure. In addition, terms related to the development of the circulatory system, including regulation of vasculature development, circulatory system process, and blood circulation were enriched among genes linked to predicted enhancer regions (e.g., *ACE*, *PDGFB*, *GATA4*, *GATA6*, *VEGFA*). This is consistent with observations that show the oral region of newborn dunnarts is highly vascularized, with blood vessels visible through their translucent skin at birth (***Cook et al., 2021***).

To gain further insight into dunnart gene regulation at this developmental stage, we scanned putative enhancers and promoters for 440 known Homer vertebrate motifs and tested for enriched TFs (***Heinz et al., 2010***). Enhancers were significantly enriched for 170 TFs relative to a background set of random GC- and length-matched sequences (FDR-corrected, p < 0.01), including those with known roles in differentiation of cranial neural crest cells (TWIST, HOXA2), skeletal morphogenesis (DLX5, CREB5, HOXA2), bone development (ATF3, RUNX), cranial nerve development (ATOH1), and/ or facial mesenchyme development (LHX2, FOXP1, MAFB; ***Figure 2B***, ***Figure 2—figure supplement***

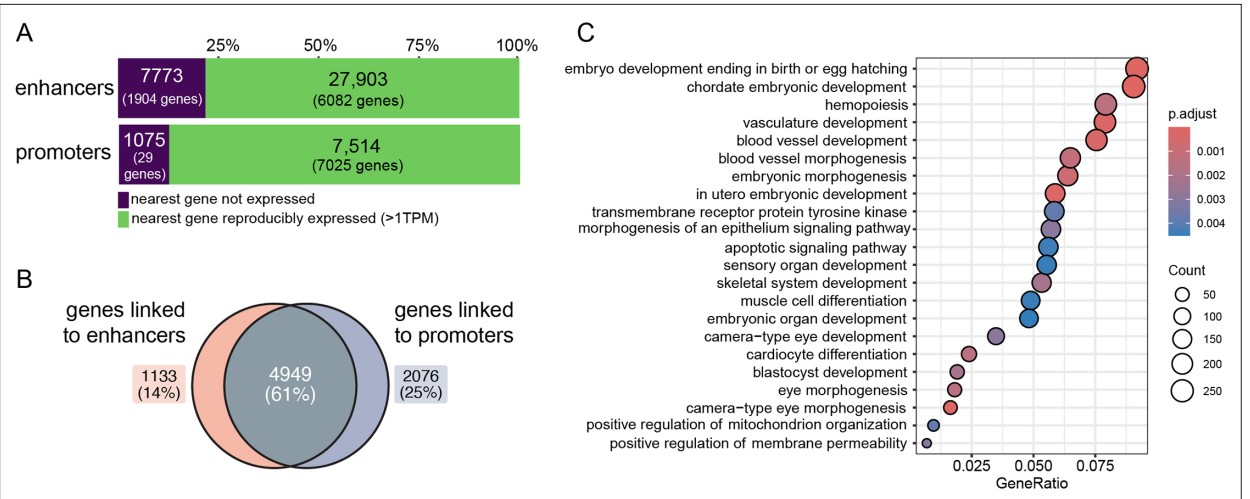

**Figure 3.** Genes linked to craniofacial enhancers and promoters in the dunnart are reproducibly expressed and involved in embryonic vasculature, muscle, skin, and sensory system development. (**A**) The majority of nearest genes assigned to candidate enhancers and promoters were reproducibly expressed in dunnart face tissue, and (**B**) reproducibly expressed genes in dunnart were associated with both a promoter and enhancer region. (**C**) Biological processes enriched for genes medium to highly expressed (>10 TPM) and linked to both a promoter and enhancer region (FDR-corrected, p < 0.01).

*1*). Consistent with the GO enrichment, TFBS in promoter sequences were dominated by transcriptional initiation regulatory sequences, with significant enrichment for 13 TFs (FDR-corrected, p < 0.01) including RFX3, RFX2, NRF, NRF1, GRY, ZBTB33, RONIN, JUND, and GFX (*Figure 2B*, *Figure 2— figure supplement 1*).

Next, we assessed predicted target gene expression by performing bulk RNA-seq in dunnart face tissue collected on the day of birth. There were 12,153 genes reproducibly expressed at a level >1 TPM across three biological replicates, with the majority of genes expressed (67%; 8158/12,153) associated with an active enhancer and/or promoter peak. Most enhancers (78% of all enhancers, *Supplementary file 1d*) or promoters (87% of all promoters, *Supplementary file 1e*) were linked to a reproducibly expressed gene in the dunnart (*Figure 3A*). Additionally, the majority of reproducibly expressed genes near a regulatory peak (61%) were associated with both an enhancer and promoter region (*Figure 3B*), highlighting the correspondence in active regulatory elements and expressed genes at this time point in the dunnart face. Genes with a medium-to-high expression level at this stage (>10 TPM) and associated with at least one putative enhancer and promoter were enriched for biological processes including 'in utero embryonic development', 'skeletal system development', 'muscle tissue development', 'skin development', 'vasculature development', and 'sensory organ development' (*Figure 3C*, *Supplementary file 1f*). Enrichment for the term 'in utero embryonic development' is indicative of the altricial nature of the dunnart neonate. In a previous study, we showed that in the dunnart, ossification begins post-birth (day of birth young corresponding approximately to embryonic day (E) 12.5 in mouse) and that the dunnart neonate instead likely relies on the well-developed cranial muscles and an extremely large chondrocranium for structural head and feeding support during early pouch life (*Cook et al., 2021*; *Clark and Smith, 1993*). Consistent with this, we observed high expression (>20 TPM) of the key head myogenesis genes (*Bentzinger et al., 2012*; *Buckingham, 2017*; *Lin et al., 2006*), *MYOD1*, *MYF6*, *MEF2C*, *PAX3*, *MYL1*, and *MYOG*, essential genes regulating chondrogenesis (*Lefebvre et al., 2019*; *Yip et al., 2019*), *SOX9*, *COL2A1*, and *FGFR1* and genes that act upstream of osteoblast differentiation (*Qi et al., 2003*; *Huang et al., 2007*), *MSX1*, *MSX2*, *CEBPA/G*, *ALPL*, *DLX3*, *DLX5*, *FGFR1*, and *FGFR2* (*Supplementary file 1d, e*).

## Comparative analyses of regulatory elements in mouse and dunnart reveal conserved and dunnart-specific enhancers during craniofacial development

Previously, we defined the postnatal craniofacial development of the dunnart and characterized the developmental differences between dunnart and mouse (*Cook et al., 2021*). Marsupials, including

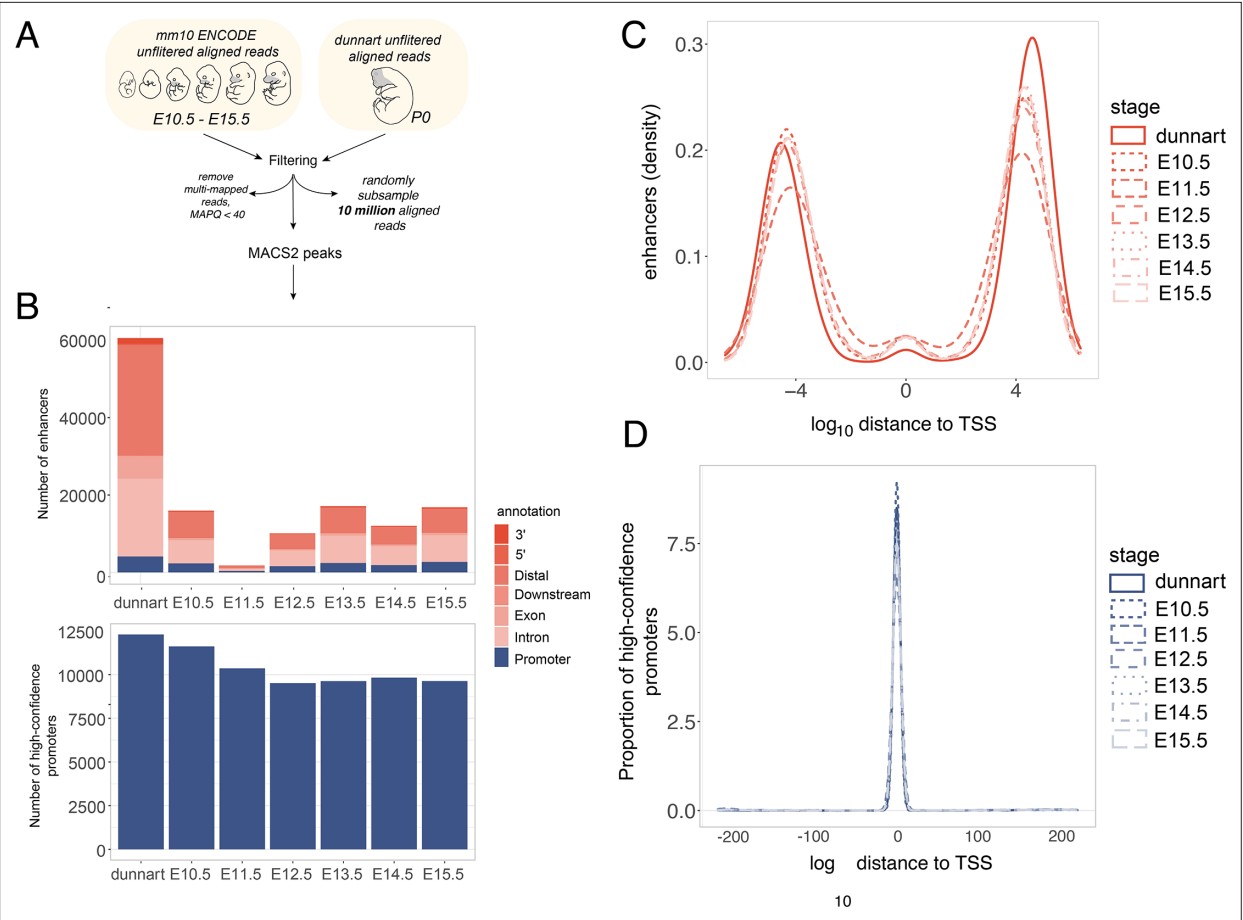

**Figure 4.** Analysis workflow and features of H3K4me3 and H3K27ac enhancers and promoters in dunnart and mouse. (**A**) Alignment filtering and peak calling workflow and (**B**) number of reproducible predicted regulatory elements identified in the dunnart and mouse embryonic stages. Log$_{10}$ of distance to the nearest TSS for putative (**C**) enhancers and (**D**) promoters.

The online version of this article includes the following figure supplement(s) for figure 4:

**Figure supplement 1.** Features of predicted promoters and enhancers in the dunnart and mouse.

dunnarts, have accelerated development of the cranial bones, musculature, and peripheral nerves when compared to placental mammals such as the mouse (*Nunn and Smith, 1998*; *Smith, 2006*; *Smith, 2001a*; *Cook et al., 2021*; *Smith, 1997*; *Veitch et al., 2000*; *Jones and Munger, 1985*; *Waite et al., 1994*). Having now characterized the chromatin landscape of the dunnart's craniofacial region, we next compared it to that of a placental mammal. From our morphological study of dunnart cranio-facial bone development, we estimated that craniofacial development in the newborn dunnart is approximately equivalent to E12.5 in mouse (*Cook et al., 2021*). However, we compared the dunnart to all available stages in the mouse to also provide insights into the regulation of additional cranio-facial features (e.g., muscle, cranial nerves, sensory system, and skin). To do this, we took advantage of publicly available craniofacial ChIP-seq data for H3K4me3 and H3K27ac generated by the mouse ENCODE consortium (*He et al., 2020*) spanning multiple developmental time points (E10.5–E15.5).

After applying consistent peak calling and filtering parameters to the dunnart ChIP-seq data, we found ~17K enhancers and ~10K promoters per stage in the mouse (*Figure 4A*, see Appendix for details). The features of mouse predicted enhancer and promoters (percentage CpG, GC content, distance from TSS, peak length, and peak intensity) were consistent with the observations in the dunnart (*Figure 4A*, *Figure 4—figure supplement 1*). After building dunnart-mm10 liftover chains (see Methods and Appendix for details), we compared mouse and dunnart regulatory elements. The alignability (conserved sequence) for dunnart enhancers to the mouse genome was ~13% for 100 bp regions (*Supplementary file 1v*). Between 0.74% and 6.77% of enhancer regions out of all alignable

enhancers were present in both mouse and dunnart (*Supplementary file 1g*). In contrast, between 45% and 57% of alignable promoter regions were present in mouse and dunnart (*Supplementary file 1g*). Although this is a small fraction of the total peaks (~8% for promoters and ~0.5% of enhancers; *Supplementary file 1g*), it suggests that, consistent with the literature (*Villar et al., 2015*), promoter regions are more stable over large evolutionary distances and that shifts in developmental timing of craniofacial marsupials may be more likely to be driven by recently evolved enhancer regions in marsupials.

To further contextualize the regulatory patterns in the dunnart compared to mouse, we retrieved mouse gene expression data from embryonic facial prominence tissue collected from E10.5 to E15.5 in the mouse ENCODE database (*He et al., 2020*), and incorporated this into our comparative analyses. We considered only one-to-one orthologous genes in mouse and dunnart. Predicted dunnart enhancers with sequence conservation in mouse and matching peak activity at any of the five mouse embryonic stages (461 regions) were located near genes reproducibly expressed (>1 TPM, 89%) in both species (*Supplementary file 1h*). Genes in this group were enriched for core developmental and signaling processes: BMP signaling, cartilage development, ossification, skeletal development, and chondrocyte differentiation (*Supplementary file 1i*). Taking the predicted dunnart enhancers alignable to the mouse but without a matching peak, we looked at nearby genes and compared expression between mouse and dunnart. For the 4311 dunnart-specific enhancers, we found that 2310 (54%) were linked to genes expressed >1 TPM in all stages and species, suggesting that these genes in mouse could be regulated by a different set of regulatory regions or could be accounted for by the reduced enrichment in the mouse ENCODE ChIP-seq face datasets for H3K27ac (*Figure 4—figure supplement 1*). We found a smaller subset (179 regions, 114 unique genes) where the nearest genes were highly expressed (>10 TPM) only in the dunnart with low to no activity in the mouse at any of the embryonic stages (E10.5–E15.5; *Supplementary file 1j*). This included genes involved in cranial neural crest proliferation and migration (*INKA2*, *TFAP2E*, *OVOL2*, *GPR161*), keratinocyte proliferation (*PLAU*, *HOXA1*), embryonic myogenesis (PDF4), development of the ectodermal placodes and sensory systems (*CNGA2*, *ELF5*, *EDN1*, *HOXA1*, *ATOH1*, *NPHP4*, *CFD*, *WNT2B*) and vomeronasal sensory neuron development (*TFAP2E*; *Supplementary file 1j*).

## Dunnart-specific expression of genes involved in the development of the mechanosensory system

Given the large evolutionary distance between the mouse and dunnart and low recovery of alignable regions, we performed a comparison between species at the gene level, by comparing genes assigned to putative enhancers and promoters between the dunnart and mouse. The number of genes that intersect can provide an idea of the similarities in genes and pathways regulated across a larger subset of the total regulatory dataset. The largest intersection size in genes with putative promoters was between the six mouse embryonic stages (1910 genes; 21.2%) and between the dunnart all six embryonic mouse stages (1908 genes, 21.2%; *Figure 5A*). Overlap between enhancers was more restricted, with 4483 predicted target genes (56%) being unique to the dunnart at D0 (*Figure 5A*). The top enriched terms for biological processes were largely shared across dunnart and mouse, with the exception of one GO term, 'sensory system development' (*Figure 5—figure supplement 1*). We further investigated this by incorporating gene expression data for mouse and dunnart for genes near putative enhancers. Genes highly expressed in dunnart but lowly or not expressed in mouse (537 total; see *Supplementary file 1k*) were related to three main developmental processes, 'epidermis/ skin development and keratinization', 'sensory system development', and 'muscle development and contraction' (*Figure 5C, D*; *Supplementary file 1l*). The majority (70/114) of genes associated with sequence conserved dunnart-specific enhancers (see *Supplementary file 1j*) overlapped with the list of genes reported here.

Genes critical for development of keratinocytes and the establishment of a skin barrier were highly expressed in dunnart facial tissue with lower expression or no transcripts expressed across the mouse embryonic stages including *IGFBP2*, *SFN*, *AQP3*, *HOPX*, *KRT17*, *KRT7*, *KRT8*, and *KRT78* (*Figure 5D*, *Supplementary file 1l*). Keratin genes are also critical for the development of the mammalian mechanosensory system (*Xiao et al., 2014*; *Doucet et al., 2013*). *Krt17*-expressing epidermal keratinocytes are necessary to establish innervation patterns during development, and *Krt8* and *Atoh1* expression is required for the specification of the Merkel cells (touch sensory cells) (*Doucet et al., 2013*).

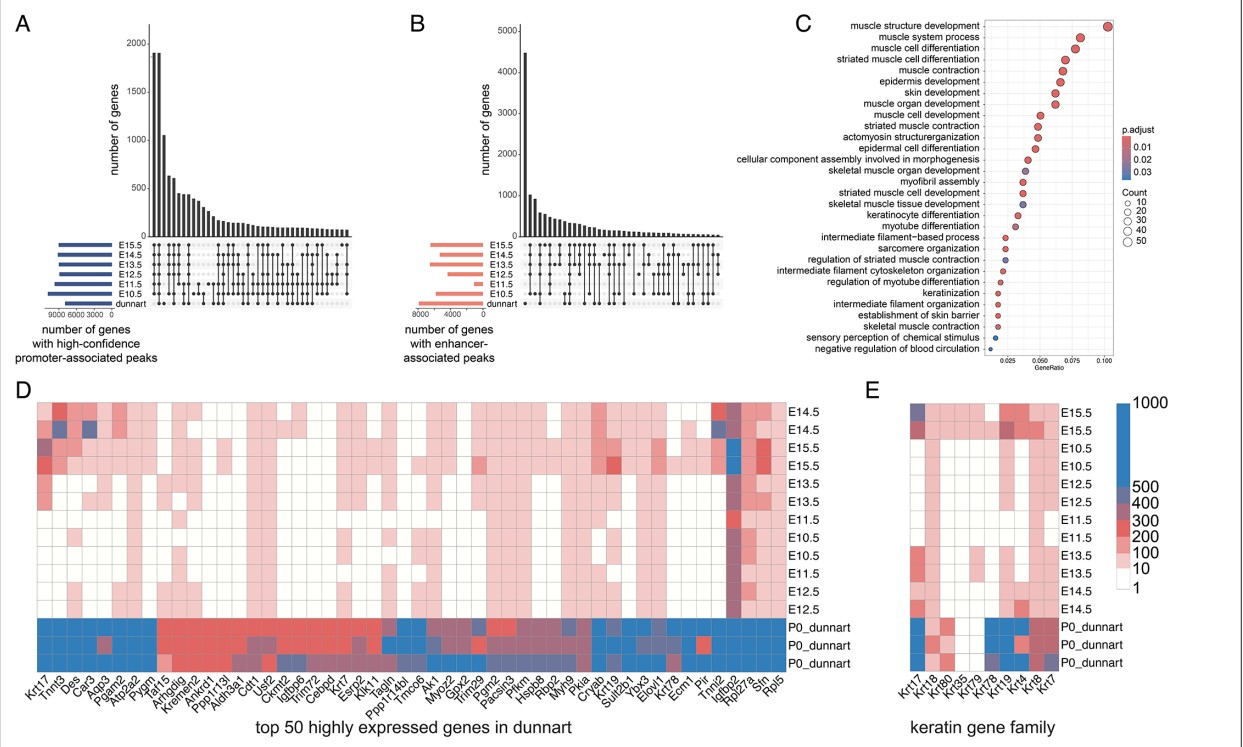

**Figure 5.** Genes near enhancers highly expressed only in dunnart are involved in the development of the skin, muscle, and mechanosensory systems. Gene set intersections across mouse (E10.5–E15.5) and dunnart (P0) for (**A**) genes near promoters and (**B**) genes near enhancers. (**C**) Gene ontology term enrichment for top 500 highly expressed dunnart genes. (**D**) Top 50 highly expressed genes (TPM) in dunnart compared with mouse embryonic stages. Scale bar in E. (**E**) Expression levels (TPM) for keratin genes across dunnart and mouse.

The online version of this article includes the following figure supplement(s) for figure 5:

**Figure supplement 1.** Top GO enriched terms in the dunnart and mouse embryonic stages for (**A**) genes near enhancers and, (**B**) genes near high-confidence promoters.

**Figure supplement 2.** Heatmaps showing gene expression values (TPM) for common developmental genes across dunnart and mouse.

**Figure supplement 3.** Volcano plot for both up- and downregulated differentially expressed genes from mouse (any stage) and dunnart.

*KRT17*, *ATOH1*, and *KRT8* are all very highly expressed in dunnart compared to the low expression in mouse facial tissue. Other genes in the keratin family such as *KRT35*, *KRT79*, *KRT4*, and *KRT80* did not vary greatly between mouse and dunnart (*Figure 5E*). Additionally, we observed high expression of genes involved in development of the olfactory system including *CCK*, *OTX1*, and *ISLR* (expressed in olfactory epithelium) (*Liu and Liu, 2018*; *Simeone, 1998*; *Santoro and Jakob, 2018*; *Haering et al., 2015*), and Ybx3 (*Haering et al., 2015*) (expressed in the nasal epithelium; *Figure 5D*, *Supplementary file 1I*). Muscle contraction genes such as *TNNI2* (*Robinson et al., 2007*) and *TNNT3* (*Wei and Jin, 2016*; *Sung et al., 2003*) and genes involved in skeletal muscle development (*Jung and Ko, 2010*; *Jin et al., 2014*; *Feng and Jin, 2016*; *Aboalola and Han, 2017*) (*CAR3, ATP2A2, IGFBP6,* and *TRIM72*) were upregulated in dunnart craniofacial tissue (*Figure 5D*, *Supplementary file 1I*). The majority of conserved toolkit genes involved in embryonic development had consistent expression across mouse and dunnart (*Figure 5—figure supplement 2*). To explore the relationship between genes expressed in the dunnart face and temporal gene expression dynamics during mouse development, we categorized mouse gene expression into five distinct temporal patterns (*Figure 6—figure supplement 1*). Each of these groups appeared to reflect biological processes occurring during development (*Figure 6—figure supplement 1*). Although dunnart facial development more closely resembles approximately E12.5 (*Cook et al., 2021*) in the mouse, when compared to the temporal gene expression dynamics during mouse craniofacial, dunnart expressed genes were associated with two distinct clusters: a set of genes upregulated specifically at E15.5 in mouse (cluster 2: OR = 1.30, CI = 1.15–1.46; *Figure 6A*; *Figure 6—figure supplement 1*) and a set of genes upregulated at E14.5

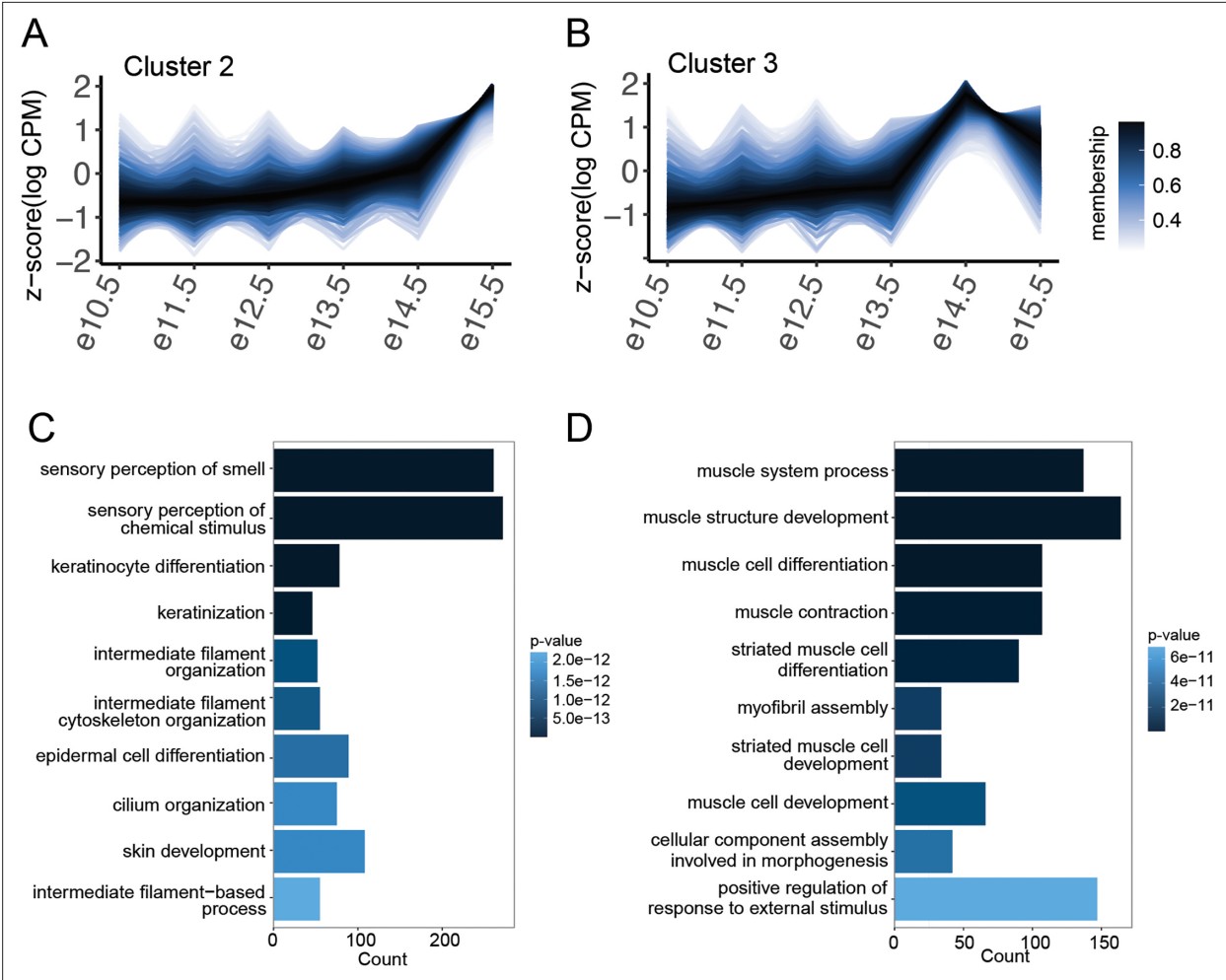

**Figure 6.** Dunnart-expressed genes are associated with two gene clusters with distinct temporal expression patterns in the mouse. (**A, B**) Genes in clusters 2 and 3 plotted with their *z*-scaled temporal expression (logCPM). Color-coding represents membership value (degree to which data points of a gene belong to the cluster). Gene ontology enrichment for biological processes enriched in (**C**) cluster 2 and (**D**) cluster 3 (FDR-corrected p < 0.01).

The online version of this article includes the following figure supplement(s) for figure 6:

**Figure supplement 1.** Temporal gene expression dynamics.

(cluster 3: OR = 1.99; CI = 1.78–2.24; *Figure 6B*; *Figure 6—figure supplement 1*). Cluster 2 genes were enriched for functions related to sensory perception, skin development, and keratinization (*Figure 6C*), while cluster 3 genes were enriched for muscle development (*Figure 6D*). These results align with our observations in dunnart enhancer and gene expression data, suggesting that shifts in the developmental timing of the skin, muscle, and sensory perception may play a role in marsupial early life history.

## Discussion

Marsupials display advanced development of the orofacial region relative to development of the central nervous system when compared to placental mammals (*Smith, 2006*; *Smith, 1997*). Specifically, the facial skeleton and muscular tissues begin development early relative to the neurocranium differentiation (*Smith, 2006*; *Smith, 1997*; *Clark and Smith, 1993*; *Smith, 1999*; *Smith, 2001b*; *Smith, 2003*; *Sánchez-Villagra and Forasiepi, 2017*). Although development of the central nervous system is protracted in marsupials compared to placentals, marsupials have well-developed peripheral motor nerves and sensory nerves (e.g., the trigeminal) at birth (*Karlen and Krubitzer, 2007*; *Smith and Keyte, 2020*). Despite increasing descriptions of craniofacial enhancers in model species

(*Attanasio et al., 2013*; *Brinkley et al., 2016*; *Samuels et al., 2020*; *Prescott et al., 2015*; *Rajderkar et al., 2024*), the genetic drivers of variation in craniofacial development between mammalian species are largely unexplored. We examined the similarities and differences in genomic regions marked by H3K4me3 and H3K27ac between the dunnart and mouse during early craniofacial development and incorporated comparisons of gene expression between species.

Given the large evolutionary distance (~170 million years) between the mouse and dunnart (*Bininda-Emonds et al., 2007*; *Luo et al., 2011*) and high turnover of non-coding DNA sequences (*Genereux et al., 2020*; *Andrews et al., 2023*), regulatory elements in the dunnart were largely not sequence conserved in the mouse (~8% for promoters and ~0.5% of enhancers). Despite this, regions with conserved activity between the two species were predominantly near genes consistently expressed in both and were enriched for core craniofacial developmental and signaling processes. Additionally, 54% of dunnart enhancers aligned to the mouse genome but lacking active marks in the mouse were associated with genes expressed throughout all mouse developmental stages, suggesting these genes might be regulated by species-specific enhancers in mice. Moreover, despite differences in experimental methods, when taking a gene-level approach, there was significant overlap in genes near regulatory elements in both species, with high concordance in enriched biological process terms. The majority of craniofacial developmental genes had highly consistent gene expression levels across the mouse and dunnart, highlighting the substantial conservation of gene regulatory networks driving facial development. This supports the notion that conserved genes and signaling pathways are crucial for mammalian facial development (*Roosenboom et al., 2016*).

Although there was generally a high level of concordance between the dunnart and mouse, we discovered dunnart-specific regulatory elements near highly expressed genes in the dunnart that were lowly or not expressed in the mouse. These genes were related to three main developmental processes, 'epidermis/skin development and keratinization', 'sensory system development', and 'muscle development and contraction'. In marsupials, pharyngeal muscles are well differentiated before birth preceding development of both the central nervous system and the skeletal system (*Smith, 1994*). Additionally, in the first 2 days immediately after birth, marsupial pouch young undergo considerable differentiation of the facial muscles (*Smith, 1994*). This pattern varies significantly from rodents where the events of skeletal and muscular development generally occur simultaneously (*Smith, 1994*). Consistent with this, we found that dunnart-expressed genes were associated with a cluster of mouse genes highly expressed at E14.5, a key time point for myogenesis in the mouse (*Stockdale, 1992*).

Our data also revealed a less evident developmental constraint experienced by marsupials, with dunnart-specific enhancer and gene expression data showing an enrichment of genes related to skin development. Newborn dunnarts exhibit high expression of KRT17 in the face and other genes essential for periderm establishment and maintenance (*Hammond et al., 2019*; *Liu et al., 2016*; *Noiret et al., 2016*; *Kousa et al., 2017*; *Carroll, 2024*; *Vaziri Sani et al., 2010*). Unlike placental mammals, marsupials retain the periderm post-birth (*Smith and Keyte, 2020*; *Lillegraven, 1975*; *Pralomkarn et al., 1990*), which may relate to neonatal transcutaneous gas exchange observed in marsupials in the first few days after birth (*Simpson et al., 2011*; *Ferner, 2018*). In placental mammals such as mouse and human, the periderm only exists in utero and disappears late in gestation with the eruption of hair and the development of a thick keratinized stratum corneum (*Hammond et al., 2019*). The persistence of the periderm aligns with the observation that dunnarts rely heavily on gas exchange through the skin and only begin lung respiration around 5 days postpartum, coinciding with the disappearance of the periderm (*Simpson et al., 2011*; *Krause et al., 1978*).

The recurrence of genes related to the development of the sensory systems in our comparative analyses spotlights a highly unique aspect of early pouch life in marsupials. Newborn marsupial young require highly developed sensory systems (mechanosensory and olfactory systems) to respond to the cues that guide them to the teat inside the mother's pouch, with the nasal-oral region making regular contact with the mother's belly during the crawl to the pouch (*Waite et al., 1994*; *Desmarais et al., 2016*; *Gemmell and Nelson, 1989*). The snout epidermis of newborn marsupials has been shown to be innervated with the presence of mature Merkel cells with connected nerve terminals (*Veitch et al., 2000*; *Jones and Munger, 1985*; *Desmarais et al., 2016*). We observe high expression of key genes involved in the development of Merkel cells and sensory placodes uniquely in the dunnart face. In the mouse, Merkel cell development does not begin until later at approximately E16.5 (*Jenkins et al.,*

*2019*). Upregulation of Merkel cell genes in dunnart facial tissue could be a result of prioritization of sensory development required for life outside the womb in a highly underdeveloped state.

In summary, our work suggests that craniofacial developmental constraints in the dunnart may be driven by dunnart-specific gene regulation. Future studies will apply single-cell multi-modal technologies to dunnart tissues to investigate the specific regulatory differences between CNS and orofacial development.

## Methods

### Tissue collection

All animal procedures, including laboratory breeding, were conducted in accordance with the current Australian Code for the Care and Use of Animals for Scientific Purposes (*NHMRC, 2013*) and were approved by The University of Melbourne Animal Ethics Committee (AEC: 1513686.2) and with the appropriate Wildlife Permit (number 10008652) from the Department of Environment, Land, Water and Planning. Animals were housed in a breeding colony in the School of BioSciences, The University of Melbourne. For details on animal husbandry and collection of dunnart pouch young, refer to *Cook et al., 2021*. Details of pouch young used in this study are presented in *Supplementary file 1m*. After removal of pouch young from the teat, young were killed by decapitation and craniofacial tissue dissected using insulin needles (Becton Dickinson). The tongue, neural tube, and eye primordia were removed to limit tissue collected to the facial prominences. Specifically, the mandibular, maxillary, and fronto-nasal prominences were collected and snap frozen in liquid nitrogen. All pouch young were determined to be <24 hr old, but to account for variability in time since birth, we scored pouch young based on head shape. Immediately after birth, the dunnart has a flat neurocranium that by approximately 1 day after birth has begun to round (see *Cook et al., 2021* for more details). We combined craniofacial tissue from 50 pouch young into two replicates, ensuring that sex, head shape, and parentage were accounted for (*Supplementary file 1m*).

### Immunofluorescence

We assessed the reactivity of the ChIP antibodies in the dunnart using immunostaining on dunnart head sections. Frontal head sections (7 μm thick) on superfrost slides (Platinum Pro, Grale) were deparaffinized and rehydrated according to standard methods (*Bancroft and Gamble, 2008*), followed by antigen retrieval using pH 8.8 unmasking solution (Vector) at 99°C for 30 min. We then incubated the sections with either rabbit anti-H3K4me3 primary antibody (1:500 dilution; Abcam ab8580) or rabbit anti-H3K27ac primary antibody (1:500, Abcam ab4729). Sections were then incubated with Alexa Fluor 555 donkey anti-rabbit antibody (1:500 dilution; Abcam in 10% horse serum in PBS with 0.1% Triton X-100; Sigma). All sections were counterstained with 300 nM DAPI to visualize cell nuclei. We observed no staining in the negative controls (no primary antibody). Images were captured using fluorescence microscopy (BX51 Microscope and DP70 Camera; Olympus) and processed in ImageJ v 2.0.0 (*Schneider et al., 2012*).

### ChIP and sequencing

ChIP was performed using the MAGnify Chromatin Immunoprecipitation System (Thermo Fisher, 492024) according to the manufacturer's instructions. Briefly, frozen dunnart tissue samples (see *Supplementary file 1m*) were diced quickly with two razor blades in cold dPBS (Gibco) followed by crosslinking in 1% formaldehyde solution (Sigma) for 10 min. We then added 0.125 M glycine and incubated for 5 min at room temperature to neutralize the formaldehyde. Chromatin was fragmented to 300 bp average size by sonication on a Covaris S2 using the following parameters: [*duty cycle = 5%, intensity = 2, cycles per burst = 200, cycle time = 60 s, cycles = 10, temperature = 4°C, power mode = frequency sweeping, degassing mode = continuous*]. In each ChIP experiment, we used sheared chromatin from each replicate for immunoprecipitation with antibodies against H3K4me3 (Abcam ab8580) and H3K27ac (Abcam ab4729). An input control was included for each replicate. DNA was purified according to kit instructions using a DynaMag-PCR Magnet (Thermo Fisher).

We assessed the success of the immunoprecipitation in dunnart craniofacial tissues by performing qPCR for primers designed to amplify genomic regions expected to be occupied by H3K4me3 and/ or H3K27ac based on mouse and human enhancers active in facial regions of E11.5 mice in VISTA

enhancer (*Visel et al., 2007*) and GeneHancer (*Fishilevich et al., 2017*; *Supplementary file 1n*). We also designed primers that amplify regions we predicted would be unoccupied by these histone modifications (enhancers active in heart tissue of E11.5 mice in VISTA enhancer browser; *Visel et al., 2007*; *Supplementary file 1n*). qPCR using SYBR Green Supermix (Bio-Rad Laboratories) was performed in triplicate on a QuantStudio 5 System (Thermo Fisher) as per the manufacturer's instructions (primers listed in *Supplementary file 1n*). The cycling conditions were [*one cycle of 95°C for 15 s, followed by 40 cycles of 95°C for 15 s, 57°C for 30 s, and 72°C for 30 s*]. A dissociation curve was also generated for each primer pair. No-template controls were included in triplicate on each plate as a negative control. Regions expected to be enriched in the test sample were quantified by expressing the test sample as a fold change relative to a control sample (no antibody control).

Illumina sequencing libraries were prepared from ChIP-enriched DNA by GENEWIZ (Suzhou, China). Libraries were constructed following the manufacturer's protocol (NEBNext UltraTMII DNA Library Prep Kit for Illumina). For each sample, a minimum of 10 ng of ChIP product was used and libraries were multiplexed and sequenced on an Illumina HiSeq 4000 instrument according to the manufacturer's instructions (Illumina, San Diego, CA, USA). Sequencing was carried out using a 2 × 150 paired-end configuration to an average depth of 57 million read pairs per sample.

## Genome assembly and annotation

In order to generate a genome for the dunnart, tissue was collected and four sequencing libraries were prepared following existing methods (*Sambrook et al., 1989*). Four libraries were generated to improve molecular complexity and genome representation of input DNA. These libraries were then sequenced using the following technologies: Illumina X Ten 2 × 150 bp, PacBio Sequel I CLR, ONT PromethION, and ONT GridION (*Supplementary file 1o*).

Quality trimming and residual adaptor removal from dunnart Illumina libraries (Library 1) was performed using trimmomatic v0.38 (*Bolger et al., 2014*) with options [*PE SLIDINGWINDOW 5:20, MINLEN 75, AVGQUAL 30*]. Contigs were assembled using 200 Gb of PacBio CLR subreads (Library 2) using Flye v2.7 (*Kolmogorov et al., 2019*) with options [*–iterations 4 –trestle –pacbio-raw –genome-size 3.0g*]. Removal of redundant contigs and two rounds of short- and long-read scaffolding were performed using Redundans 0.14a (*Pryszcz and Gabaldón, 2016*) with options [*--nogapclosing* `--limit 1.0`]. Inputs for redundancy scaffolding were short-insert paired-end reads (Library 1) and 6.5 gigabases of Oxford Nanotechnology reads corresponding to two libraries (Library 3 and Library 4). Scaffolds then underwent two rounds of polishing to improve base quality using Pilon v1.23 (*Walker et al., 2014*) with Illumina Library 1 as input and with options [*--vcf* `--diploid` `--chunksize 10000000` `--fix snps`,*indexls,gaps* `--minqual 15`]. The resulting genome assembly had a total size of 2.84 Gb and an N50 length of 23 megabases (Mb). We used BUSCO v5.2.2 to assess genome completeness [*v3.0.2, -l mammalia_odb10 -m genome*]. BUSCO gene recovery was 89.9% for complete orthologs in the mammalia_odb10 lineage dataset which includes 9226 BUSCOs. Together, these metrics indicate that the assembly is of comparable completeness and contiguity to other recently published marsupial genomes (*Feigin et al., 2018*; *Feigin et al., 2022*; *Renfree et al., 2011*; *Murchison et al., 2012*; *Mikkelsen et al., 2007*; *Peel et al., 2021*; *Brandies et al., 2020*) and therefore represents an excellent resource for downstream functional genomic experiments. The resulting de novo assembly with a resulting scaffold N50 of 23 Mb and a total size of 2.84 Gb makes it comparable to other marsupial genomes (*Feigin et al., 2018*; *Renfree et al., 2011*; *Murchison et al., 2012*; *Mikkelsen et al., 2007*). The G+C content of the de novo contigs in the dunnart (~36.25%) was similar to other marsupial species Tasmanian devil: 36.4% (*Murchison et al., 2012*), thylacine: ~36% (*Feigin et al., 2018*), wallaby: 34–38.8% (*Renfree et al., 2011*), opossum: 37.8% (*Mikkelsen et al., 2007*), woylie: 38.6% (*Peel et al., 2021*), and the brown antechinus: 36.2% (*Brandies et al., 2020*).

Gene annotations from the high-quality genome assembly of the Tasmanian devil (*Sarcophilus harrisii*, GCF_902635505.1 – mSarHar1.11), which has a divergence time with the dunnart of approximately 40 million years, were accessed from NCBI and then lifted over to dunnart scaffolds using the program liftoff v1.0 (*Shumate and Salzberg, 2021*) with option [*--d 4*].

## Whole-genome alignment

To compute pairwise genome alignments between the mouse and dunnart, we used the mouse mm10 assembly as the reference. We first built pairwise alignments using Lastz and axtChain to generate

co-linear alignment chains (*Kent et al., 2003*), using the previously described Lastz parameters for vertebrates, [*K = 2400, L = 3000, Y = 3400, H = 200*] with the HoxD55 scoring matrix (*Sharma and Hiller, 2017*). After building chains, patchChain (*Sharma and Hiller, 2017*) was applied to extract all the unaligned loci and create local alignment jobs for each one. The new local alignments were combined with the original local alignments for an improved set of chains. We then applied chain-Cleaner (*Suarez et al., 2017*) with the parameters [*-LRfoldThreshold = 2.5 -doPairs -LRfoldThreshold-Pairs = 10 -maxPairDistance = 10,000 -maxSuspectScore = 100,000 -minBrokenChainScore = 75,000*] to improve the specificity of the alignment. After generating an improved set of chains, we applied chainPreNet, chainNet, and ChainSubset to filter, produce the alignment nets and create a single chain file using only the chains that appear in the alignment nets (*Kent et al., 2003*). Alignment nets are a hierarchical collection of the chains that attempt to capture orthologous alignments (*Kent et al., 2003*). Chain fragments were joined using chainStitchId and dunnart to mouse chains generated using chainSwap (*Kent et al., 2003*). For quality control, maf files were generated using netToAxt and axtToMaf (*Kent et al., 2003*). Block counts, block lengths, and pairwise divergence in the alignments were assessed using MafFilter (*Dutheil et al., 2014*).

## ChIP sequencing data analysis

First, we assessed the raw sequencing read quality using FastQC v0.11.9 (*Andrews, 2010*). Raw data were processed by adapter trimming and low-quality read removal using Cutadapt v1.9.1 (*Martin, 2011*) [*-q 20a AGATCGGAAGAGCACACGTCTGAACTCCAGTCA -A AGATCGGAAGAGCGTCGTGT AGGGAAAGAGTGT* --max-n *0.10 -m 75*]. ChIP sequencing statistics for raw reads and trimmed reads are described in *Supplementary file 1p*. Sequencing reads were aligned to the dunnart genome with Bowtie2 v.2.3.5.1 (*Langmead and Salzberg, 2012*) [*-q -X 2000* --very-sensitive]. Unfiltered aligned reads from ChIP-seq experiments performed using mouse embryonic facial prominence for E10.5, E11.5, E12.5, E13.5, E14.5, and E15.5 were downloaded from https://www.encode.org/ (*He et al., 2020*) (accession details described in *Supplementary file 1q*). For both dunnart and mouse-aligned reads, low-quality and unpaired reads were removed using Samtools v.1.9 (*Li et al., 2009*) [*-q 30 -f 2*] and duplicate reads removed by the MarkDuplicates tool from picard v.2.23.1 (https://broadinstitute.github.io/picard/). Mapping statistics and library complexity for dunnart and mouse reads are described in *Supplementary file 1r and s*, respectively. Effective genome size for the dunnart was calculated using the [*unique-kmers.py*] script from khmer v.2.0 (*Crusoe et al., 2015*).

Peaks were called on the dunnart-aligned reads using MACS2 v.2.1.1 (*Zhang et al., 2008*) [*-f BAMPE*] and the mouse aligned reads using MACS2 v.2.1.1 with default parameters for mm10, using total DNA input as control and retaining all statistically significant enrichment regions (FDR-corrected $p < 0.01$). Reproducible consensus peaks for biological replicates within a species were determined using the ENCODE3 [*overlap_peaks.py*] script (*He et al., 2020*). Enriched regions were considered reproducible when they overlapped in two biological replicates within a species by a minimum of 50% of their length using bedtools intersect v2.29.2 (*Quinlan and Hall, 2010*). Peak-calling statistics for dunnart and mouse are described in *Supplementary file 1t and u*, respectively. Similar to *Villar et al., 2015*, we overlapped H3K4me3 and H3K27ac reproducible peaks to determine promoter-associated peaks (marked by only H3K4me3 or both H3K4me3 and H3K27ac) and enhancer-associated peaks (marked only by H3K27ac). H3K4me3 and H3K27ac reproducible peaks were overlapped to determine genomic regions enriched for H3K4me3, H3K27ac, or both marks using bedtools intersect v2.29.2 (*Quinlan and Hall, 2010*). Double-marked H3K4me3 and H3K27ac elements were defined as regions reproducibly marked by H3K4me3 and H3K27ac and overlapping by a minimum 50% of their reciprocal length and were merged with bedtools v2.29.2 (*Quinlan and Hall, 2010*). Mouse and dunnart peaks called on aligned reads are deposited at 10.7554/eLife.103592.1.

## Gene ontology enrichment analyses

For the dunnart peaks, gene annotations lifted over from the Tasmanian devil annotation were associated with ChIP-seq peaks using the default settings for the annotatePeak function in ChIPseeker v1.26.2 (*Yu et al., 2015*). As there is no equivalent gene ontology database for dunnart, we converted the Tasmanian devil RefSeq IDs to Ensembl v103 IDs using biomaRt v2.46.3 (*Durinck et al., 2009*; *Durinck et al., 2005*), and then converted these to mouse Ensembl v103. In this way, we were able to use the mouse Ensembl gene ontology annotations for the dunnart gene domains. We were able

to assign Devil Ensembl IDs to 74% of genes with peaks, and mouse IDs to 95% of genes with a devil Ensembl ID. For calculating enrichment in GO in the dunnart, the list of Tasmanian devil genes with an orthologous Ensembl gene in the mouse was used as the background list. All gene ontology analyses were performed using clusterProfiler v4.1.4 (*Wu et al., 2021*), with gene ontology from the org. Mm.eg.db v3.12.0 database (*Carlson, 2020*), setting an FDR-corrected p-value threshold of 0.01 for statistical significance.

Short sequence motifs enriched in dunnart peaks were identified with Homer v4.11.1 (*Heinz et al., 2010*) using [*findMotifsGenome.pl*]. In this case, random GC- and length-match sequences for all promoters and enhancers were used as the background set to test for enrichment compared to random expectation. Enriched motifs were clustered into Homer motif families (*Heinz et al., 2010*).

## Mouse and dunnart peak comparison

Dunnart and mouse peaks called from normalized input reads were filtered to 100-bp regions centered on the peak summit. Dunnart peak coordinates were lifted over using liftOver (*Kent et al., 2003*) to the mouse (mm10) genome and then back to the dunnart genome. Alignable peaks were kept if after reciprocal liftOver they had the same nearest gene call (*Supplementary file 1v*). Alignable peaks were then intersected with enhancer and promoters at each stage in the mouse with bedtools intersect v.2.30.0 (*Quinlan and Hall, 2010*) to assess peaks with conserved activity (*Supplementary file 1w*).

## RNA-sequencing and analyses

The reads for three replicates were generated from facial prominences of P0 dunnart pouch young. Each replicate is a pool of two pouch young. Libraries were prepared using the Illumina stranded mRNA kit and were sequenced to a depth ≥50 M read pairs (i.e., 25 M F + 25 M R) in 2 × 100 bp format. The raw reads were first trimmed using Trimmomatic v0.39 (*Bolger et al., 2014*) [ILLUMINACLIP:2:30:10 SLIDINGWINDOW:5:15 MINLEN:50 AVGQUAL:20]. Reads were mapped with hisat2 v2.21 (*Zhang et al., 2021*) [--fr --no-mixed --no-discordant] and alignments were filtered using samtools view (*Li et al., 2009*) [-f 1 -F 2316]. Finally, the count table was generated using featureCounts from the Subread package v2.0.1 (*Liao et al., 2013*) [-s 2 -p -t exon --minOverlap 10]. Mouse gene count tables were acquired from ENCODE (see *Supplementary file 1q*). Mouse and dunnart gene expression data were normalized for library size with edgeR v.4.0.16 (*Robinson et al., 2010*) and lowly expressed genes filtered (>2 cpm) for at least two out of three replicates.

For mouse gene expression data, the resulting gene list was tested for temporally, differentially expressed genes with the [*DBanalysis*] function in the TCseq package (*Wu and Gu, 2020*), which implements edgeR to fit read counts to a negative binomial generalized linear model. Differentially expressed genes with an absolute $\log_2$ fold change >2 compared to the starting time point (E10.5) were determined. The optimal division of clusters was determined using the Calinski criterion implemented with the [*cascadeKM*] function in the vegan v. package (*Oksanen, 2020*) with the parameters [*inf.gr = 1, sup.g = 10 iter = 1000, criterion = 'calinski'*]. Time-clustering for five clusters was performed using the [*timeclust*] function with the parameters: [*algo = 'cm', k = 5, standardize = TRUE*] which performs unsupervised soft clustering of gene expression patterns into five clusters with similar $z$-scaled temporal patterns. Orthologous genes reproducibly expressed >1 TPM in the dunnart were compared to the list of genes for each cluster using Fisher's exact test followed by p-value corrections for multiple testing with the Holm method. Gene ontology enrichment was performed using [*enrichGO*] as part of the clusterProfiler v4.10.1 (*Wu et al., 2021*) with the 9933 genes present across the clusters as background and a false discovery rate of 1%.

## Acknowledgements

St Vincent's Institute acknowledges the infrastructure support it receives from the National Health and Medical Research Council Independent Research Institutes Infrastructure Support Program and from the Victorian Government through its Operational Infrastructure Support Program. The authors thank Eva Suric, Tania Long, and Darren Cipolla for their technical contributions and help with animal husbandry.

## Additional information

### Competing interests

Andrew J Pask: is funded in part by Colossal Biosciences. However, Colossal Biosciences had no direct influence on this project design, data collection, analysis, interpretation, or conclusions presented in this paper. The other authors declare that no competing interests exist.

### Funding

| Funder | Grant reference number | Author |
| --- | --- | --- |
| Australian Government | Research Training Program Scholarship | Laura E Cook |
| Australian Research Council | DP160103683 | Andrew J Pask |

The funders had no role in study design, data collection, and interpretation, or the decision to submit the work for publication.

### Author contributions

Laura E Cook, Conceptualization, Data curation, Formal analysis, Investigation, Visualization, Methodology, Writing – original draft, Writing – review and editing; Charles Y Feigin, Formal analysis, Investigation, Writing – review and editing; John D Hills, Investigation; Davide M Vespasiani, Formal analysis; Andrew J Pask, Conceptualization, Resources, Supervision, Funding acquisition, Writing – review and editing; Irene Gallego Romero, Resources, Supervision, Funding acquisition, Writing – original draft, Writing – review and editing

### Author ORCIDs

Laura E Cook ⓘ https://orcid.org/0000-0002-4459-2592
Charles Y Feigin ⓘ https://orcid.org/0000-0003-4981-5254
Irene Gallego Romero ⓘ https://orcid.org/0000-0003-1613-8998

### Ethics

All animal procedures, including laboratory breeding, were conducted in accordance with the current Australian Code for the Care and Use of Animals for Scientific Purposes and were approved by The University of Melbourne Animal Ethics Committee (AEC: 1513686.2) and with the appropriate Wildlife Permit (number 10008652) from the Department of Environment, Land, Water and Planning.

Reviewer #1 (Public review): https://doi.org/10.7554/eLife.103592.3.sa1
Author response https://doi.org/10.7554/eLife.103592.3.sa2

## Additional files

### Supplementary files

Supplementary file 1. All supplementary tables. (a) Annotated dunnart peak nearest gene ID conversions. (b) Enriched gene ontology terms for dunnart genes near promoters. (c) Enriched gene ontology terms for dunnart genes near enhancers. (d) Dunnart enhancers and nearest gene expression (TPM). (e) Dunnart promoters and nearest gene expression (TPM). (f) Enriched GO terms for dunnart enhancers and promoters where nearest gene expression ≥10 TPM (FDR <1%). (g) Summary of mouse–dunnart conserved peak activity for a. high-confidence putative promoters and b. putative enhancers. (h) Enhancers with conserved activity in mouse and dunnart with incorporated stage and species gene expression values (TPM). (i) Gene ontology terms for biological processes enriched in genes expressed in mouse and dunnart (>1 TPM) near conserved enhancers. (j) Dunnart-specific enhancers where the nearest genes were highly expressed (>10 TPM) only in the dunnart with low to no activity in the mouse at any of the embryonic stages (E10.5–E15.5). (k) Highly expressed genes (TPM) in dunnart where expression is lower or absent in mouse. (l) Biological processes enrichment for highly expressed genes in dunnart where expression is lower or absent in mouse. (m) Dunnart facial prominence tissue samples. (n) qPCR primer sequences for ChIP enrichment validation. (o) Details of libraries used in genome assembly. (p) DNA fragments

isolated after pull down were sequenced by GENEWIZ. Paired-end sequencing at an average depth of 57 million reads. a. Raw reads sequencing statistics, b. Read statistics after CutAdapt filtering. (q) ENCODE mouse embryonic facial prominence ChIP-seq and gene expression quantification file accession numbers and details used in study. (r) Read alignment quality metrics for H3K4me3 and H3K27ac dunnart ChIP-seq samples. (s) Read alignment quality metrics for mouse ENCODE ChIP-seq data. (t) Peak calling quality metrics for H3K4me3 and H3K27ac dunnart ChIP-seq samples. (u) Peak calling quality metrics for mouse ChIP-seq peaks after peak calling with MACS2. (v) Summary of alignable enhancer-associated peaks in the mouse and dunnart. (w) Summary of alignable high-confidence promoter-associated peaks in mouse and dunnart.

MDAR checklist

### Data availability

The data generated in this study have been deposited in NCBI's Gene Expression Omnibus (*Barrett et al., 2013*) and are available through GEO Series accession number GSE188990. Accession IDs for previously published datasets from the ENCODE consortium (*He et al., 2020*) are given in *Supplementary file 1*. Processed data and an IGV session for browsing are available in a figshare repository. All analyses described were carried out using custom bash, Python3, and R v4.1.0 scripts and are available at https://github.com/lecook/chipseq-cross-species (copy archived at *Cook, 2025*).

The following datasets were generated:

| Author(s) | Year | Dataset title | Dataset URL | Database and Identifier |
|---|---|---|---|---|
| Cook LE, Feigin CY, Hills JD, Vespasiani DM, Pask AJ, Gallego Romero I | 2023 | dunnart genome | https://doi.org/10.6084/m9.figshare.22089080 | figshare, 10.6084/m9.figshare.22089080 |
| Cook LE, Feigin CY, Hills JD, Vespasiani DM, Pask AJ, Gallego Romero I | 2023 | Gene regulatory dynamics during craniofacial development in a carnivorous marsupial | https://www.ncbi.nlm.nih.gov/geo/query/acc.cgi?acc=GSE188990 | NCBI Gene Expression Omnibus, GSE188990 |
| Cook LE, Feigin CY, Hills JD, Vespasiani DM, Pask AJ, Gallego Romero I | 2023 | dunnart and mouse peak characterisation | https://doi.org/10.6084/m9.figshare.21748058 | figshare, 10.6084/m9.figshare.21748058 |
| Cook LE, Feigin CY, Hills JD, Vespasiani DM, Pask AJ, Gallego Romero I | 2023 | tc-seq processed data | https://doi.org/10.6084/m9.figshare.21747806 | figshare, 10.6084/m9.figshare.21747806 |
| Cook LE, Feigin CY, Hills JD, Vespasiani DM, Pask AJ, Gallego Romero I | 2023 | liftover chains | https://doi.org/10.6084/m9.figshare.21905904 | figshare, 10.6084/m9.figshare.21905904 |

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

## Appendix 1

### Validation of ChIP-seq in dunnart craniofacial tissue

As the H3K4me3 and H3K27ac antibodies used in this work had not been previously used in marsupials, we first tested their reactivity in the dunnart using immunofluorescence and observed strong positive staining for both antibodies (*Appendix 1—figure 1*). We validated the levels of enrichment in the ChIP libraries using qPCR and primers for sequences in the dunnart that were orthologous to the craniofacial enhancers mm428, mm387, mm423, hs466 and GH07J005561 and an enhancer active in the heart as a negative control, hs222 (*Supplementary file 1n*). All primer sets were enriched as a percentage of the input control, and as expected, no enrichment was observed in the rabbit and mouse mock IgG negative control samples (*Appendix 1—figure 1*). Although the dunnart ortholog of the human heart enhancer (hs222) was also enriched in both ChIP samples, this enhancer has only been tested at E11.5 in mouse embryos (*Visel et al., 2007*) and may have alternative roles at other stages in development or in different species.

Having validated the ability of our antibodies to enrich for dunnart chromatin marks, ChIP-seq libraries were sequenced to average depth of 57 million reads. Reads were mapped to a de novo assembly of the dunnart genome generated for this study (*Supplementary file 1o*), which we annotated using the publicly available high-quality genome assembly of the Tasmanian devil (mSarHar1.11; see Methods for additional details). After mapping and filtering, we retained an average of 43 million mapped reads per library (*Supplementary file 1p*). All samples had an NRF of between 0.76 and 0.85 (*Supplementary file 1r*; *He et al., 2020*). Read coverage was highly correlated for replicates within each mark (Pearson's $r > 0.99$) and also between marks (Pearson's $r > 0.88$; *Appendix 1—figure 2*). Read coverage for replicate input controls was also correlated, but as expected, due to enrichment for specific genomic regions in IP samples, there was no correlation in read coverage observed between the input controls and IP samples (Pearson's $r$ range = 0.03–0.08, *Appendix 1—figure 2*). Similarly, alignment BAM files for H3K4me3 and H3K27ac IP replicates show strong and specific enrichment indicated by a prominent and steep rise of the cumulative sum toward the highest rank, whereas input control alignment BAM files are roughly diagonal, highlighting even coverage across the genomic windows (*Appendix 1—figure 2*). Fraction of reads in peaks was approximately 60% for consensus H3K27ac and H3K4me3 peaks (*Supplementary file 1t*).

### Investigating promoters located >3 kb from an annotated TSS

To investigate the large number of putative promoters located >3 kb from an annotated TSS, we first looked at the relationship between the number of peaks per gene and the distance to the next closest gene. After annotating peaks with the nearest gene call using ChIPseeker (*Yu et al., 2015*), 72% of genes had at least 1 predicted promoter or enhancer, with 81 genes having more than 50 promoters (*Appendix 1—figure 3*). The number of promoters per gene was weakly positively correlated with distance to the next closest gene (Pearson's $r = 0.36$, $p < 2.2 \times 10^{-16}$), suggesting that this observation might be partially due to the presence of unannotated transcripts between genes (*Appendix 1—figure 3*). Enhancers can also be associated with H3K4me3 (*Wu and Gu, 2020*). However, enrichment levels (peak intensity) tend to be lower than that of H3K27ac (*Kim et al., 2010*). Hence, peaks characterized as promoters that were located more than 3 kb from a known TSS may instead represent active enhancer regions (*Kim et al., 2010*). To assess this, we compared mean peak intensity in H3K4me3 peaks located greater than 3 kb from the nearest TSS to H3K4me3 peaks located within 3 kb of the TSS. We found that H3K4me3 peaks located closer to the TSS had a stronger peak signal (mean = 46.10) than distal H3K4me3 peaks (mean = 6.95; Wilcoxon FDR-adjusted $p < 2.2 \times 10^{-16}$). This suggests that although some distal promoter peaks may be due to missingness in the annotation, the majority likely represent peaks associated with enhancer regions.

### Defining high-confidence promoter regions

This analysis identified three clusters: upstream distal peaks (cluster 3, 31,285 peaks), downstream distal peaks (cluster 2, 22,863 peaks), and peaks centered on the TSS (cluster 1, 12,295 peaks; *Appendix 1—figure 4*). Cluster 1 peaks had a higher intensity (*Appendix 1—figure 4*) and length (*Appendix 1—figure 4*) when compared to clusters 2 and 3 (Wilcoxon FDR-adjusted $p < 2.2 \times 10^{-16}$) and compared to all predicted enhancers and promoters (Wilcoxon FDR-adjusted $p < 2.2 \times 10^{-16}$). Furthermore, peaks in cluster 1 (those closest to annotated TSS) had a higher GC and CpG island content than clusters 2 and 3 (Wilcoxon FDR-adjusted $p < 2.2 \times 10^{-16}$) and higher than enhancers (Wilcoxon FDR-adjusted $p < 2.2 \times 10^{-16}$), consistent with known features of mammalian promoters

(*Appendix 1—figure 4*). H3K4me3 ChIP-seq peak intensity has previously been correlated with transcriptional activity (*Barski et al., 2007*) and a higher peak intensity has also been observed in promoters in mammalian liver tissue (*Villar et al., 2015*). We thus used cluster 1 (12,295 peaks) to define a set of high-confidence promoters for all of the following analyses.

## Investigating differences in the number of mouse and dunnart peaks called

After applying consistent peak calling and filtering parameters to the dunnart ChIP-seq data, we found that the number of peaks was fairly consistent across mouse embryonic stages (~20–30,000 total peaks; *Figure 4A*). This number was significantly lower than the total number of peaks observed in the dunnart (~150,000 peaks; *Figure 4A*). We investigated this further and found that the strength and specificity of enrichment differed between the mouse and dunnart datasets. Dunnart alignment BAM files for H3K4me3 and H3K27ac immunoprecipitation replicates show strong and specific enrichment (*Appendix 1—figure 2*); however, mouse alignment BAM files show a weaker enrichment with the immunoprecipitation samples for H3K27ac being closer to the input (*Appendix 1—figure 5*). This was also reflected in the similarity between mouse IPs and input controls based on read coverage, with correlation coefficients higher (H3K4me3, Pearson's *r* range = 0.17–0.32, *Appendix 1—figure 6*; H3K27ac, Pearson's *r* range = 0.39–0.64, *Appendix 1—figure 6*) than the corresponding dunnart values (*Appendix 1—figure 2*). Therefore, this is likely a technical confounder that may be responsible for the lower numbers of enriched regions called in the mouse, as peaks with lower enrichment signal might not be identified by the peak caller. This was consistent with the distribution of peak enrichment values in the dunnart and mouse for H3K4me3, with a lower mean peak enrichment value in the dunnart compared to all mouse stages (Kruskal–Wallis FDR-corrected p = 6.7 × 10$^{-12}$; *Appendix 1—figure 2*). However, this was not the case for peak enrichment values for H3K27ac peaks (*Appendix 1—figure 2*), which generally have lower peak enrichment values than observed in H3K4me3 (*Villar et al., 2015*; *Kim et al., 2010*)

## Whole-genome alignment between mouse and dunnart

In order to compare activity in orthologous mouse and dunnart peaks, liftOver chains are needed to find the corresponding coordinates in the alternative genome. After building a dunnart-mm10 liftOver chain, we examined the quality of the WGA between mouse and dunnart. Approximately a quarter of both genomes was recovered in the WGA, with 26% of the dunnart genome and 28% of the mouse genome present with an average mismatch between the dunnart and mouse of 37.2% (*Appendix 1—figure 7*). Previous WGAs between placentals and marsupials (opossum vs. human, tammar wallaby vs. human) have recovered approximately 6–8% of the genomes with LastZ (*Howe et al., 2021*). The higher recovery in our WGA may be due to the use of new genome alignment tools like chainCleaner and patchChain. Given the higher conservation of coding regions than non-coding regions, exon coverage is a useful metric for assessing the quality of the alignment. 92% of mouse exon sequence and 69% of dunnart exon sequence was present in the WGA (*Appendix 1—figure 7*). Coverage of the exons recovered after liftOver was 82.7% and 57.8% in the mouse and dunnart, respectively (*Appendix 1—figure 7*). The lower exon coverage in the dunnart could be due to the missingness in the dunnart annotation. The majority of the alignment blocks fell within the 100 bp to 1000 bp size range with a total of 578.3 Mb of DNA sequence alignment out of a total of 801 Mb in the alignment within this range (*Appendix 1—figure 7*). This is very similar to the existing opossum-human WGA where 83% of alignment blocks fall within the size range of 100–1000 bp (*Howe et al., 2021*).

We then used the liftOver chains to find corresponding coordinates for dunnart peaks in the mouse genome. Given the majority of the alignment blocks for the mouse and dunnart WGA fell between 100 and 1000 bp in length (*Appendix 1—figure 7*), we first assessed the impact of peak length (centered on the peak summit, which represents the point with the highest signal in the peak region) on interspecies recovery of peaks. We defined dunnart peak regions as alignable if they could be reciprocally lifted over to the mouse genome and then back to the dunnart genome. Unsurprisingly, for both enhancers and promoters, the number of alignable peaks recovered decreased with increasing peak length (*Supplementary file 1v and w*). Promoters had the highest number of regions alignable (17%) using 50 bp peak summit widths, which decreased to 1.8% with a 500 bp peak summit length (*Supplementary file 1w*). Similarly, 19% of enhancers were alignable using a 50-bp peak summit length, and this decreased to 2% with 500 bp peak summit length (*Supplementary file 1v*). To assess the accuracy of the reciprocal liftOver, we next investigated

whether alignable peaks in the mouse and dunnart mapped to the same nearest Ensembl gene before and after liftOver. Accuracy was high for both the promoters and enhancers (from 94% to 100%; *Supplementary file 1v, w*).

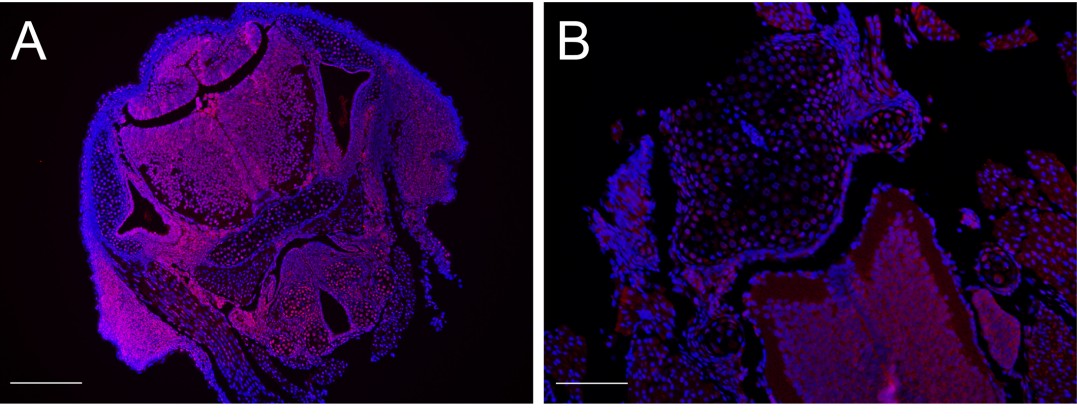

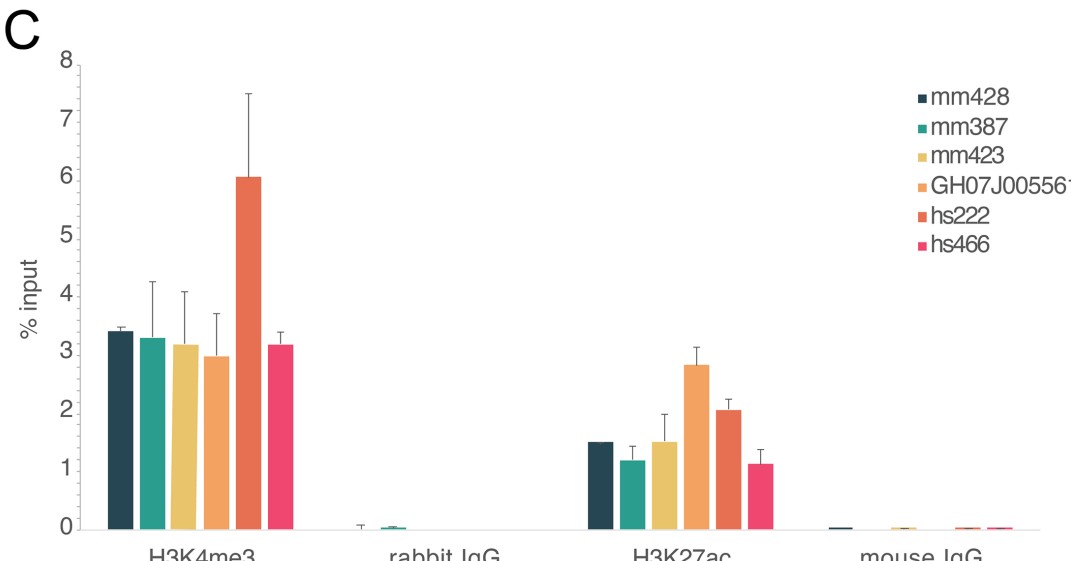

**Appendix 1—figure 1.** Validation of antibodies in dunnart craniofacial tissue with immunofluorescence and qPCR. Localization of (**A**) H3K27ac (pink, scale bar = 250 µm) and (**B**) H3K4me3 (pink, scale bar = 80 µm), in D0 dunnart head sections with nuclei stained with DAPI (blue) (**C**) enhancer regions expected to be enriched in the IP samples presented as the percentage of the input control sample as measured by qPCR.

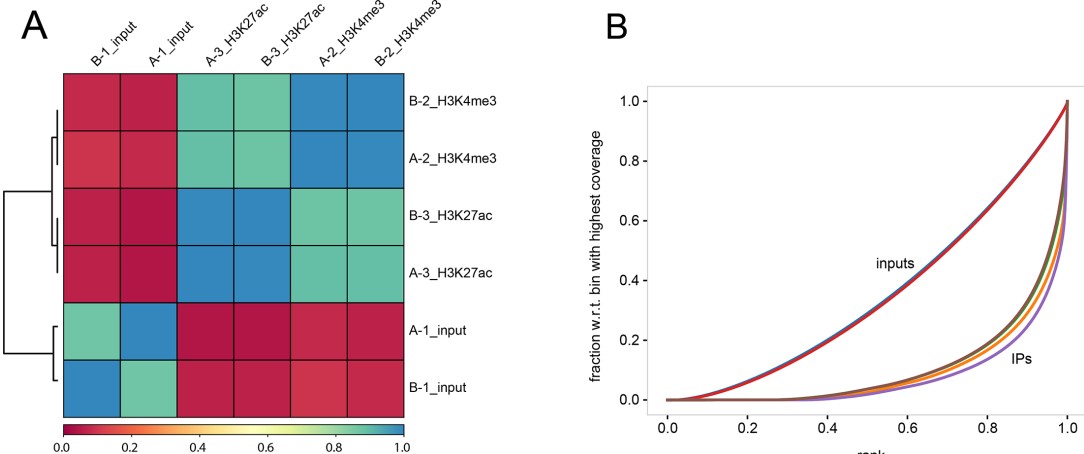

**Appendix 1—figure 2.** deepTools quality control plots for dunnart subsampled aligned BAM files. (**A**). Overall similarity between BAM files based on read coverage within genomic regions with Pearson correlation coefficients plotted for H3K27ac, H3K4me3, and input control. (**B**) Fingerprint plot showing a profile of cumulative read coverages for each BAM file. All reads overlapping a window (bin) of the specified length are counted, sorted, and plotted for H3K27ac, H3K4me3, and input control.

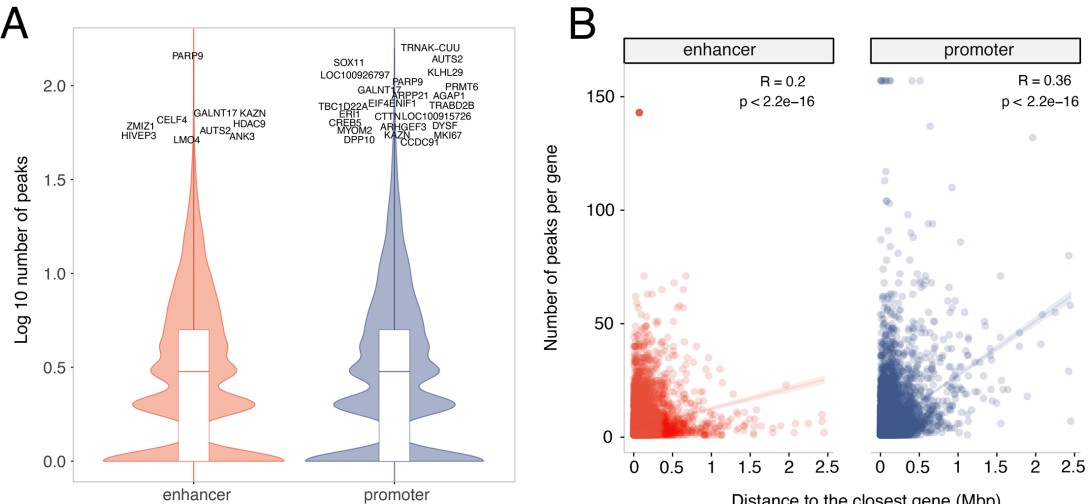

**Appendix 1—figure 3.** Number of peaks per gene for enhancer- and promoter-associated peaks. (**A**) $Log_{10}$ number of peaks per gene. Genes with greater than 50 peaks are noted. (**B**) Scatter plot with distance to the closest gene on the x-axis and number of peaks per gene on the y-axis. There is a weak but significant correlation between the number of peaks per gene and the distance to the next closest gene in both enhancers and promoters.

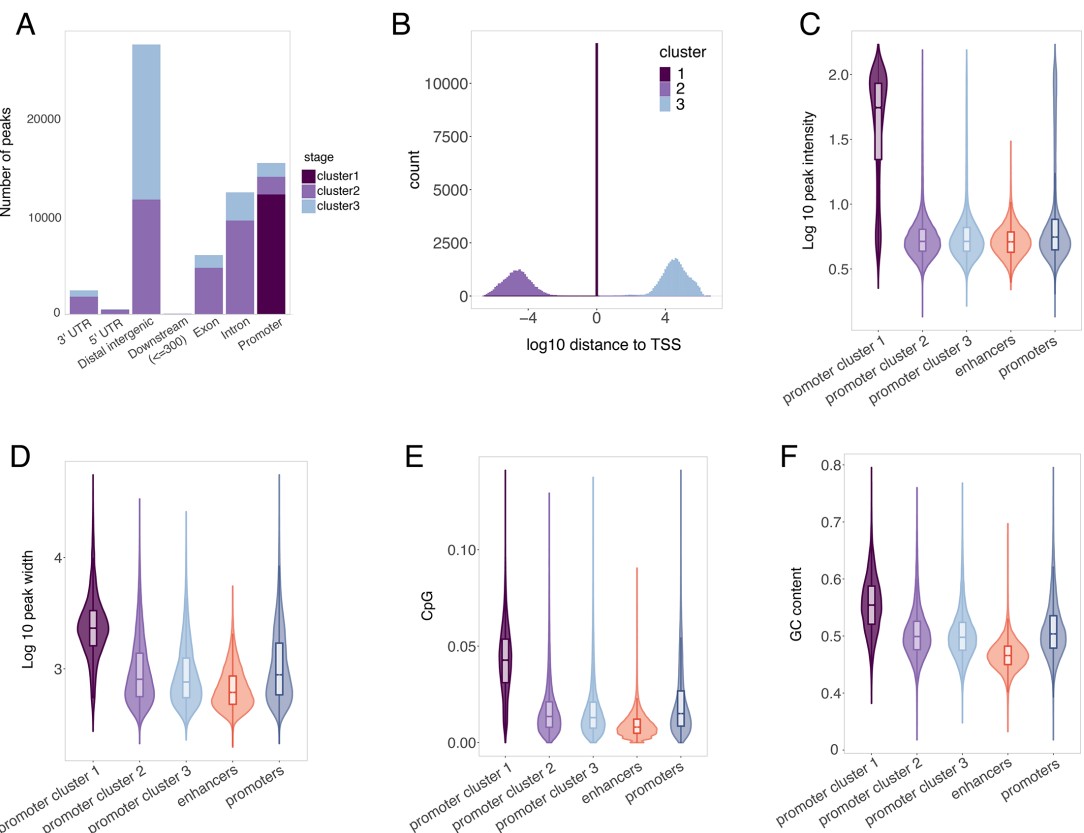

**Appendix 1—figure 4.** Subsetting high-confidence promoter-associated peaks with *k*-means clustering.
(**A**) Barplot for the number of promoter-associated peaks per cluster and histogram showing the distribution of
peak distance from the nearest TSS in each cluster. (**B**) Genomic annotations for promoter-associated peaks in each
cluster. (**C**) GC content, (**D**) Log$_{10}$ peak length, (**E**) CpG content, and (**F**) Log$_{10}$ of clustered promoter-associated
peaks, enhancer-associated peaks, and unclustered promoter-associated peaks.

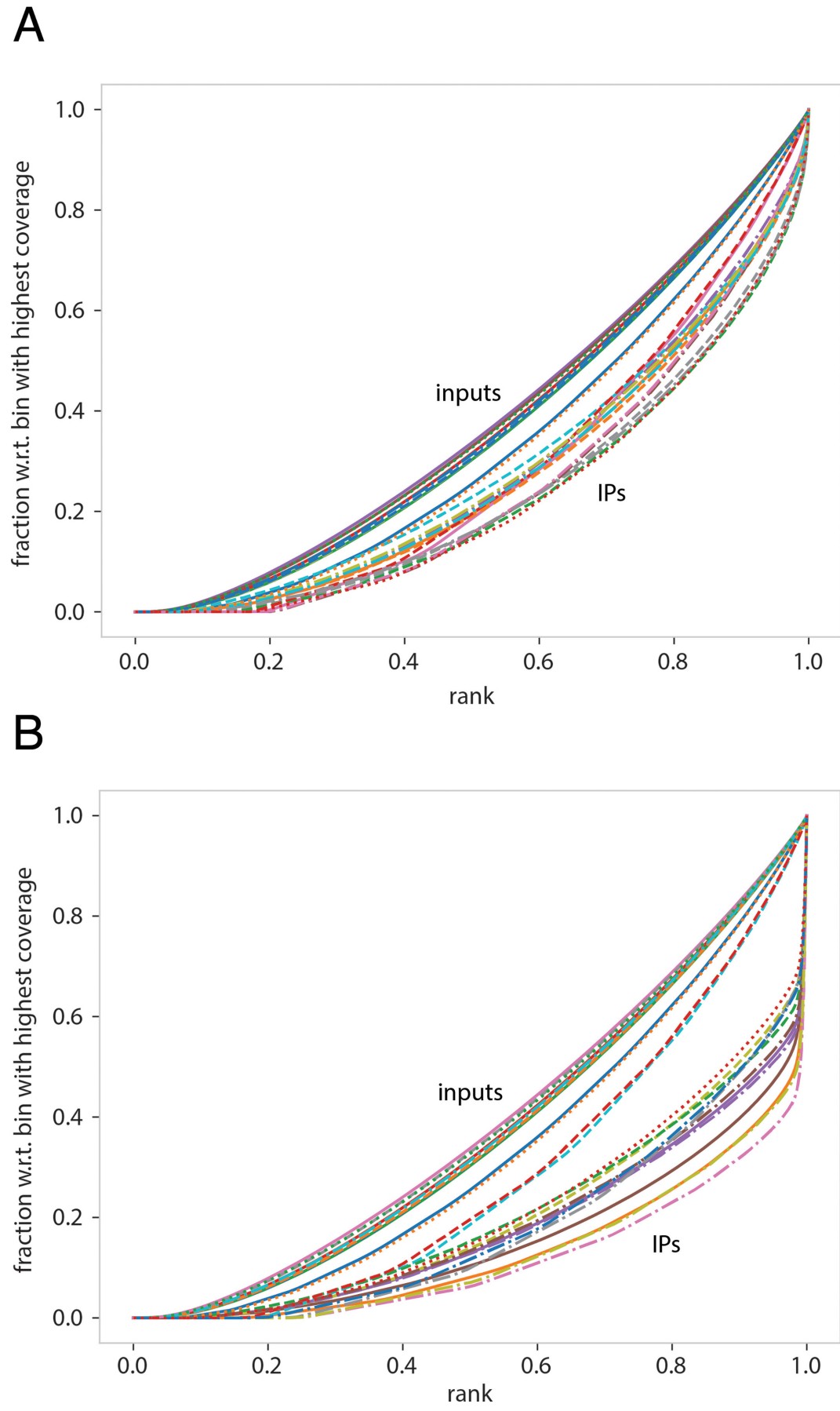

**Appendix 1—figure 5.** deepTools fingerprint plot for mouse subsampled aligned BAM files. Cumulative read coverages for each BAM file. All reads overlapping a window (bin) of the specified length are counted, sorted, and plotted for (**A**) H3K27ac and (**B**) H3K4me3.

A

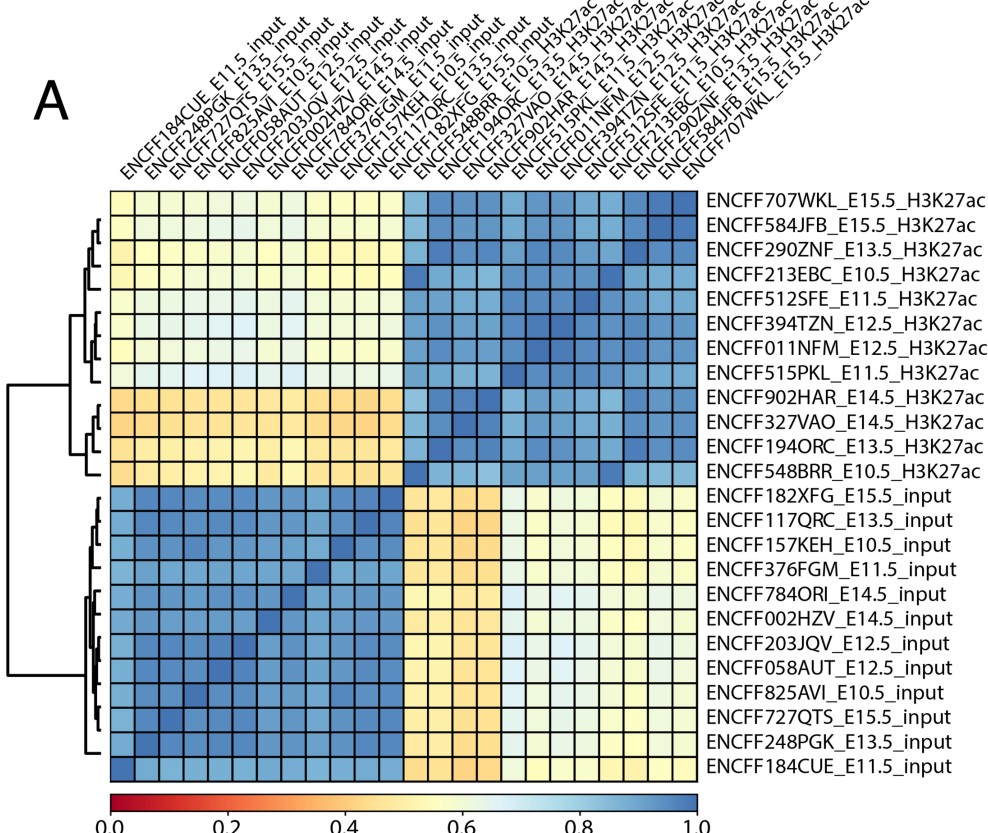

B

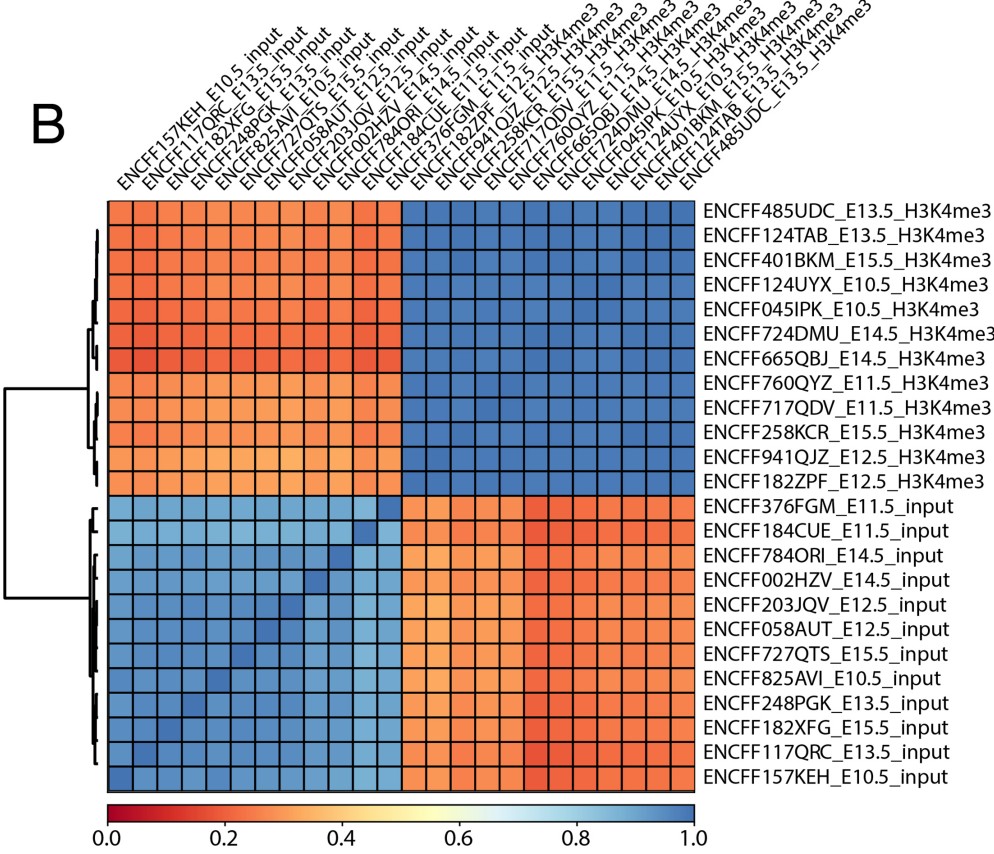

**Appendix 1—figure 6.** deepTools correlation plots for mouse subsampled aligned BAM files. Overall similarity between BAM files based on read coverage within genomic regions with Pearson correlation coefficients plotted for (**A**) H3K27ac and (**B**) H3K4me3.

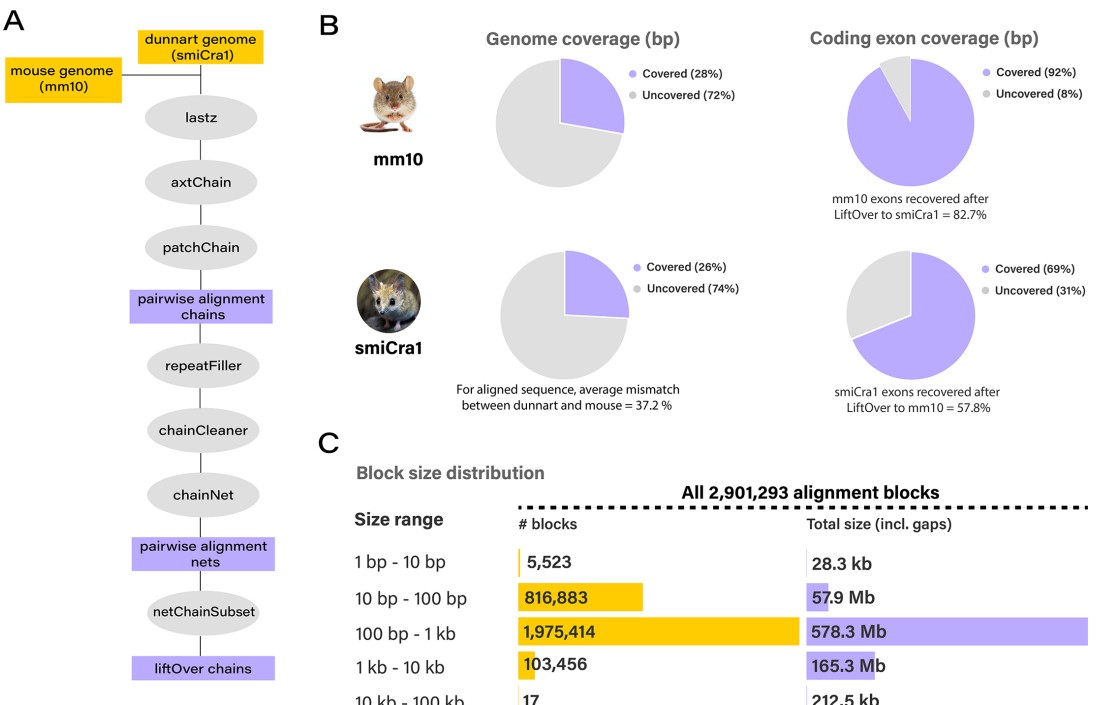

**Appendix 1—figure 7.** Genome alignment between the fat-tailed dunnart *Sminthopsis crassicaudata* and mouse (*Mus musculus*). (**A**) Genome alignment workflow. (**B**) Genome coverage and exon coverage (bp) between mouse and dunnart. Coverage of exons recovered after LiftOver. (**C**) Block size distribution for the dunnart including number of blocks and total size of the blocks.

