## [Editor Report · eLife Assessment]

This **important** study of regulatory elements and gene expression in the craniofacial region of the fat-tailed dunnart shows that, compared to placental mammals, marsupial craniofacial tissue develops in a precocious manner, with enhancer regulatory elements as primary driver of this difference. The **compelling** data, including a new dunnart genome assembly, provide an invaluable reference for future mammalian evolution studies, especially once additional developmental time point for the fat-tailed dunnart become available.

---

## [Referee Report · Reviewer #1 (Public review)]

Summary:

Compared to placental mammals, marsupials have a short gestation period and give birth to altricial young. To assist with the detection and response to cues that direct the neonate joeys to the mother's pouch, as well as latching onto the teat, marsupial craniofacial development at this stage is rapid and heterochronous relative to placentals. Cook et al. have presented an important study on the transcriptomic and epigenomic signature underlying this heterochronous development of craniofacial features across mammals, using the fat-tailed dunnart as a marsupial model.

Given the lack of a dunnart genome, the authors prepared long and short read sequence datasets to assemble and annotate a novel genome to allow for mapping of RNAseq and ChIPseq data against H3K4me3 and H3K27ac, which allowed for identification of putative promoter and enhancer sites in dunnart. They found that genes proximal to these regulatory loci were enriched for functions related to bone, skin, muscle and embryonic development, verifying the precocious state of newborn dunnart facial tissue. When compared with mouse, the authors found a much higher proportion of promoter regions aligned between species than for enhancer regions, and subsequent profiling identified regulatory elements conserved across species and are important for mammalian craniofacial development. In contrast, identification of dunnart-specific enhancers and patterns of RNA expression further confirm the precocious state of muscle development, as well as for sensory system development, in dunnart, suggesting that early formation of these features are critical for neonate marsupials.

Marsupials are emerging as an important model for studying mammalian development and evolution, and the authors have performed a novel and thorough analysis that helps to elucidate the regulatory profile underlying craniofacial heterochrony. Impressively, this study also includes the assembly of a new marsupial reference genome that will benefit many future studies of mammalian developmental biology.

Strengths:

The genome assembly method was thorough, using two different long-read methods (Pacbio and ONT) to generate the long reads for contig and scaffold construction, increasing the quality of the final assembled genome, which was effectively annotated and used for functional analysis of orthologous regulatory elements.

The birth of altricial young in marsupials is an important feature of their development that is distinct from placental mammals which are separated by about 160 million years of evolution. Very little is known, however, about the regulatory profile that contributes to the advanced craniofacial development required for joey survival. This is one of the few epigenomic studies performed in marsupials (of any organ) and the first performed in fat-tailed dunnart (also of any organ), which begins to address this lack of knowledge.

The study also provides evidence supporting the important role enhancer elements play in mammalian phenotypic evolution, relative to promoters.

Weaknesses:

Biological replicates of facial tissue were collected at a single developmental time point of the fat-tailed dunnart within the first postnatal day (P0), and analysed this in the context of similar mouse facial samples from the ENCODE consortium at six developmental time points, where previous work from the authors have shown that the younger mouse samples (E11.5-12.5) approximately corresponds to the dunnart developmental stage (Cook et al. 2021). However, it would be useful to have samples from at least one older dunnart time point, for example, at a developmental stage equivalent to mouse E15.5. This would provide additional insight into the extent of accelerated face development in dunnart relative to mouse, i.e. how long do the regulatory elements that are activated early in dunnart remain active for and does their function later influence other aspects of craniofacial development?

---

## [Author Response]

The following is the authors’ response to the original reviews.

**Reviewer #1 (Public review):**
Summary:Cook et al. have presented an important study on the transcriptomic and epigenomic signature underlying craniofacial development in marsupials. Given the lack of a dunnart genome, the authors also prepared long and short-read sequence datasets to assemble and annotate a novel genome to allow for the mapping of RNAseq and ChIPseq data against H3K4me3 and H3K27ac, which allowed for the identification of putative promoter and enhancer sites in dunnart. They found that genes proximal to these regulatory loci were enriched for functions related to bone, skin, muscle and embryonic development, highlighting the precocious state of newborn dunnart facial tissue. When compared with mouse, the authors found a much higher proportion of promoter regions aligned between species than for enhancer regions, and subsequent profiling identified regulatory elements conserved across species and are important for mammalian craniofacial development. In contrast, the identification of dunnart-specific enhancers and patterns of RNA expression further confirm the precocious state of muscle development, as well as for sensory system development, in dunnart suggesting that early formation of these features are critical for neonate marsupials likely to assist with detecting and responding to cues that direct the joeys to the mother's teat after birth. This is one of the few epigenomic studies performed in marsupials (of any organ) and the first performed in fat-tailed dunnart (also of any organ). Marsupials are emerging as an important model for studying mammalian development and evolution and the authors have performed a novel and thorough analysis, impressively including the assembly of a new marsupial reference genome that will benefit many future studies.Strengths:The study provides multiple pieces of evidence supporting the important role enhancer elements play in mammalian phenotypic evolution, namely the finding of a lower proportion of peaks present in both dunnart and mouse for enhancers than for promoters, and dunnart showing more genes uniquely associated with it's active enhancers than any other combination of mouse and dunnart samples, whereas this pattern was less pronounced than for promoter-associated genes. In addition, rigorous parameters were used for the cross-species analyses to identify the conserved regulatory elements and the dunnart-specific enhancers. For example, for the results presented in Figure 1, I agree that it is a little surprising that the average promoter-TSS distance is greater than that for enhancers, but that this could be related to the possible presence of unannotated transcripts between genes. The authors addressed this well by examining the distribution of promoter-TSS distances and using proximal promoters (cluster #1) as high confidence promoters for downstream analyses.The genome assembly method was thorough, using two different long read methods (Pacbio and ONT) to generate the long reads for contig and scaffold construction, increasing the quality of the final assembled genome.Weaknesses:Biological replicates of facial tissue were collected at a single developmental time point of the fat-tailed dunnart within the first postnatal day (P0), and analysed this in the context of similar mouse facial samples from the ENCODE consortium at six developmental time points, where previous work from the authors have shown that the younger mouse samples (E11.5-12.5) approximately corresponds to the dunnart developmental stage (Cook et al. 2021). However, it would be useful to have samples from at least one older dunnart time point, for example, at a developmental stage equivalent to mouse E15.5. This would provide additional insight into the extent of accelerated face development in dunnart relative to mouse, i.e. how long do the regulatory elements that activated early in dunnart remain active for and does their function later influence other aspects of craniofacial development?

We thank the reviewer for their feedback and agree that the inclusion of multiple postnatal stages in the dunnart would give further valuable insights to the comparative analyses. Unfortunately, we were limited by the pouch young available and prioritized ensuring robust data at a single stage for this study. We hope to expand this work to more stages in future studies.

The authors refer to the development of the CNS being delayed in marsupials relative to placental mammals, however, evidence shows how development of the dunnart brain (whole brain or cortex) is protracted compared to mouse, by a factor of at least 2 times, rather than delayed per se (Workman et al. 2013; Paolino et al. 2023). In addition, there is evidence that cortical formation and cell birth may begin at approximately the same stage across species equivalent to the neonate period in dunnart (E10.5 in mouse), and that shortly after this at the stage equivalent to mouse E12.5, the dunnart cortex shows signs of advanced neurogenesis followed by a protracted phase of neuronal maturation (Paolino et al. 2023). Therefore, it is possible that marsupial CNS development appears delayed relative to mouse but instead begins at the same stage and then proceeds to develop on a different timing scale.

The comparison here is not directly between CNS development in placental and marsupials but CNS development relative to development of a subset of structures of the cranial skeleton and musculature (as first proposed by Kathleen Smith 1997). For example, Smith 1997 found that in eutherians, evagination of the telencephalon and appearance of the pigment in the eye occur before the ossification of the premaxilla, maxilla, and dentary. However, in marsupials, evagination of the telencephalon and appearance of the pigment in the eye occur concurrently with condensation of cartilage in the basicranium and the ossification of the premaxilla, maxilla, and dentary. Smith 1997 reports both a delay in the initiation of CNS development in marsupials relative to craniofacial ossification and a protraction of CNS development compared to placental mammals.

This also highlights the challenges of correlating different staging systems between placentals and marsupials as stages determined as equivalent can change depending on which developmental events are used. The protracted development of the CNS in marsupials (Smith 1997, Workman et al. 2013; Paolino et al. 2023) still supports the hypothesis that during the short gestation period in marsupials structures required for life outside the womb in an embryonic-like state, such as the orofacial region, are likely prioritized.

We have clarified this based on the reviewers feedback and added text referring to the protraction of marsupial CNS development to the Discussion section.

[New text]: Marsupials display advanced development of the orofacial region relative to development of the central nervous system when compared to placental mammals[3,6].

[New text]: Although development of the central nervous system is protracted in marsupials compared to placentals, marsupials have well-developed peripheral motor nerves and sensory nerves (eg. the trigeminal) at birth [5].

**Reviewer #2 (Public review):**
This study by Cook and colleagues utilizes genomic techniques to examine gene regulation in the craniofacial region of the fat-tailed dunnart at perinatal stages. Their goal is to understand how accelerated craniofacial development is achieved in marsupials compared to placental mammals.The authors employ state-of-the-art genomic techniques, including ChIP-seq, transcriptomics, and high-quality genome assembly, to explore how accelerated craniofacial development is achieved in marsupials compared to placental mammals. This work addresses an important biological question and contributes a valuable dataset to the field of comparative developmental biology. The study represents a commendable effort to expand our understanding of marsupial development, a group often underrepresented in genomic studies.The dunnart's unique biology, characterized by a short gestation and rapid craniofacial development, provides a powerful model for examining developmental timing and gene regulation. The authors successfully identified putative regulatory elements in dunnart facial tissue and linked them to genes involved in key developmental processes such as muscle, skin, bone, and blood formation. Comparative analyses between dunnart and mouse chromatin landscapes suggest intriguing differences in deployment of regulatory elements and gene expression patterns.Strengths(1) The authors employ a broad range of cutting-edge genomic tools to tackle a challenging model organism. The data generated - particularly ChIP-seq and RNA-seq from craniofacial tissue - are a valuable resource for the community, which can be employed for comparative studies. The use of multiple histone marks in the ChIP-seq experiments also adds to the utility of the datasets.(2) Marsupial occupy an important phylogenetic position, but they remain an understudied group. By focusing on the dunnart, this study addresses a significant gap in our understanding of mammalian development and evolution. Obtaining enough biological specimens for these experiments studies was likely a big challenge that the authors were able to overcome.(3) The comparison of enhancer landscapes and transcriptomes between dunnarts and can serve as the basis of subsequent studies that will examine the mechanisms of developmental timing shifts. The authors also carried out liftover analyses to identify orthologous enhancers and promoters in mice and dunnart.Weaknesses and Recommendations(1) The absence of genome browser tracks for ChIP-seq data makes it difficult to assess the quality of the datasets, including peak resolution and signal-to-noise ratios. Including browser tracks would significantly strengthen the paper by provide further support for adequate data quality.

We have put together an IGV session with the dunnart genome, annotation and ChIP-seq tracks. This is now available in the FigShare data repository (10.7554/eLife.103592.1).

(2) The first two figures of the paper heavily rely in gene orthology analysis, motif enrichment, etc, to describe the genomic data generated from the dunnart. The main point of these figures is to demonstrate that the authors are capturing the epigenetic signature of the craniofacial region, but this is not clearly supported in the results. The manuscript should directly state what these analyses aim to accomplish - and provide statistical tests that strengthen confidence on the quality of the datasets.

As this is the first epigenomic profiling for this species we performed extensive data quality control (See Supplementary Tables 2-3, 18, 20-23 and Supplementary Figures 1-3, 6-11). These figures and corresponding Supplementary Tables show the robustness of the data, including well-described metrics for assessing promoters and enhancers, GO terms relevant to craniofacial development and binding motifs for key developmental TF families.

We have emphasised this aspect of the work more strongly in the results section, particularly in [Defining craniofacial putative enhancer- and promoter regions in the dunnart].

(3) The observation that "promoters are located on average 106 kb from the nearest TSS" raises significant concerns about the quality of the ChIP-seq data and/or genome annotation. The results and supplemental information suggest a combination of factors, including unannotated transcripts and enhancer-associated H3K4me3 peaks - but this issue is not fully resolved in the manuscript. The authors should confirm that this is not caused by spurious peaks in the CHIP-seq analysis - and possibly improve genome annotation with the transcriptomic datasets presented in the study.

Spurious ChIP-seq peaks could be possible as there is no “blacklisted regions” database for the dunnart to filter on, however we used a no-IP control, a stringent FDR of 0.01 and peaks had to be reproducible in two biological replicates when calling peaks - all of which should reduce the likelihood of false positives.

H3K4me3 activity at enhancers is well-established, in particular when enhancer sequences are also bound by RNA Pol II (Koch and Andrau, 2011; Pekowska et al., 2011). However, compared to H3K4me3 activity at promoters, H3K4me3 levels at enhancers are low (Calo and Wysocka, 2013). This is in line with our observations that H3K4me3 levels at enhancers are much lower than observed at promoter regions (see Supplementary Note 2). We found that H3K4me3 peaks located closer to the TSS had a stronger peak signal (mean = 46.10) than distal H3K4me3 peaks (mean = 6.95; Wilcoxon FDR-adjusted p < 2.2 x 10^-16^). This suggests that although some distal promoter peaks may be due to missingness in the annotation, the majority likely represent peaks associated with enhancer regions. We have emphasized this finding more strongly in the results section:

[New text]: H3K4me3 activity at enhancers is well-established[25,26], however, compared to H3K4me3 activity at promoters, H3K4me3 levels at enhancers are low[27]. This is in line with our observations where H3K4me3 levels at distal enhancer peaks are nearly 7 times lower than those observed at promoter regions (see SupNote2).

(4) The comparison of gene regulation between a single dunnart stage (P1) and multiple mouse stages lacks proper benchmarking. Morphological and gene expression comparisons should be integrated to identify equivalent developmental stages. This "alignment" is essential for interpreting observed differences as true heterochrony rather than intrinsic regulatory differences.

Given the developmental differences between eutherian and marsupial mammals it is challenging to assign the dunnart a precise “equivalent” developmental stage to the mouse. From our morphological and developmental characterisation (see Cook et al. 2020 Nat Comms Bio) based on ossification patterns the dunnart orofacial region on the day of birth appears to be similar to that of an E12.5 mouse embryo (just prior to the observation of ossified craniofacial bones). However, when we compared both regulatory elements and expressed genes between the dunnart at this stage (P1) and 5 developmental stages in the mouse, there is no obvious equivalent stage. For example, when we simply compare genes linked to enhancer peaks, the group with the largest intersection between dunnart and any mouse stage are ~500 genes that are present in dunnart, and mouse stages E10.5, E12.5 - E15.5, Figure 5B. When we then compare genes expressed in the dunnart to temporal gene expression dynamics during mouse development we find that the largest overlap is with genes highly expressed at E14.5 or E15.5 in the mouse (Figure 6, Supplementary Figure 5). We have strengthened the rationale for the selected mouse stages in the comparative analyses section of the results.

(5) The low conservation of putative enhancers between mouse and dunnart (0.74-6.77%) is surprising given previous reports of higher tissue-specific enhancer conservation across mammals. The authors should address whether this low conservation reflects genuine biological divergence or methodological artifacts (e.g., peak-calling parameters or genome quality). Comparisons with published studies could contextualize these findings.

The reported range (0.74 - 6.77%) refers to the number regions called as an active enhancer peak in both species (conserved activity) divided by the total number of dunnart peaks alignable to the mouse genome, which we expect to be low given sequence turnover rates and the evolutionary distance separating dunnart and mice. The alignability (conserved sequence) for dunnart enhancers to the mouse genome was ~13% for 100bp regions and can be found in Supplementary Table 22, we have now clarified this in the main text.

[New Text]: After building dunnart-mm10 liftover chains (see Methods and SupNote5) we compared mouse and dunnart regulatory elements. The alignability (conserved sequence) for dunnart enhancers to the mouse genome was ~13% for 100bp regions (Supplementary Table 22).

The activity conservation range reported here is consistent with previously reported for marsupial-placental enhancer comparisons (Villar et al. 2015), where ~1% of conserved liver-specific human enhancers had conserved activity to opossum. Follow up studies in Berthelot et al 2018 also found that approximately 1% of human liver enhancers were conserved across the placental mammals included in the study.

(6) Focusing only on genes associated with shared enhancers excludes potentially relevant genes without clear regulatory conservation. A broader analysis incorporating all orthologous genes may reveal additional insights into craniofacial heterochrony.

We appreciate the reviewers comment, we understand that a broader analysis may provide some additional insights to this question however in this study our focus was understanding the enhancers driving craniofacial development in these species. We linked enhancers with gene expression data as additional evidence of regulatory programs involved in craniofacial development. The majority (~70%) of genes reproducibly expressed were linked to an active enhancer and/or promoter. This has now been highlighted in the result section.

[New Text]: There were 12,153 genes reproducibly expressed at a level > 1 TPM across three biological replicates, with the majority of genes 67% of genes expressed (67%; 8158/12153) associated with near an active enhancer and/or promoter peak.

In conclusion, this study provides an important dataset for understanding marsupial craniofacial development and highlights the potential of genomic approaches in non-traditional model organisms. However, methodological limitations, including incomplete genome annotation and lack of developmental benchmarking weaken the robustness and of the findings. Addressing these issues would significantly enhance the study's utility to the field and its ability to support the study's central conclusion that dunnart-specific enhancers drive accelerated craniofacial development.
**Reviewer #1 (Recommendations for the authors):**
Minor comments and corrections:(1) ChIP-seq FRiP fractions were much higher in dunnart samples than in mouse. Is this related to any differences in sample preparation they are aware of in the ENCODE datasets of mouse, such as different anti-histone antibodies used (and therefore different efficiency of binding to the same histone markers across species)? The authors appear to have addressed something similar with respect to the much lower enriched peak number observed in the mouse sample relative to dunnart in Supp note 4. I suspect the "technical cofounder" they refer to there is affecting both the FRiP scores and the higher correlation coefficients between IP and input in mouse.

We chose the same antibodies used in the mouse craniofacial tissue ENCODE experiments however, the procedure is slightly different. We used the MAGnify Chromatin Immunoprecipitation System while in the ENCODE assays performed by Bing Ren’s group in 2012 was an in-house lab protocol for MicroChIP. Given that the samples for mouse and dunnart were not processed together, by the same researcher, with the same protocol there could be any number of technical cofounders impacting enrichment. A low FRiP score suggests low specificity as the majority of reads are in non-specific regions (low enrichment), consistent with the higher correlation between IP and input in mouse. The data quality also appears to vary between H3K27ac and H3K4me3 in the mouse (Supplementary Table 21), with H3K4me3 FRiP scores more similar to those observed in our dunnart experiments. This suggests a potential confounder specific to the mouse H3K27ac IP. QC metrics (FRiP, bam correlation) are consistent between H3K27ac and H3K4me3 IPs in our experiments (Supplementary Table 20).

(2) Some of the promoter peak numbers in Supp table 1 do not match the numbers in the main text.

We have corrected the incorrect number reported in the text for promoter peaks with orthologous genes (8590 -> 8597).

(3) In Supp tables 2 and 3, the number of GO terms similar across tables is 466, which is ~42% of total number of enriched GO terms. However the authors mention that only 23% of terms were the same between promoters and enhancers, and a value of 42% was applied to the proportion of terms uniquely enriched for terms associated with genes assigned to promoters only. Unless I'm reading these Supp tables incorrectly, is it possible the proportions were mixed up?

Thanks for catching this. The lists provided in Supplementary Table 2 were incorrect. The Supplementary Tables and in text description has been corrected to reflect this.

(4) Would be helpful to add a legend for the mouse samples in Supp Figure 10.

We have added the labels to the plot.

(5) In Supp note 5, regarding the percentage of alignable peaks recovered, the percentages mentioned for the 50bp and 500bp peak summit lengths for enhancers and promoters do not seem to match the values in Supp tables 22 and 23.

Thank you for catching this - we have corrected the Supplementary Tables and in text.

(6) Please provide additional information to explain how dunnart RNA expression was associated with the five temporal expression clusters found in the mouse data shown in Figure 6 given there is only one dunnart time point and so the species temporal pattern's could not be compared, i.e. how was the odds ratio calculated and was this applied iteratively for dunnart against each mouse age and within each temporal cluster?

The TCseq package takes the mouse expression data across all 6 stages and calls differentially expressed genes with an absolute log_2_ fold-change > 2 compared to the starting time-point (E10.5). The mouse gene expression patterns were clustered into 5 clusters that each show distinct temporal expression patterns (see Supplementary Figure 5D). The output from this is 5 lists where within each list are unique genes that share a temporal pattern. These lists of mouse genes were then each compared to the orthologous genes expressed in the dunnart using a Fishers Exact test with corrections for multiple testing using the Holm method. We have added additional details in the methods:

[New text]: Orthologous genes reproducibly expressed >1 TPM in the dunnart were compared to the list of genes for each cluster using Fisher’s Exact Test followed by p-value corrections for multiple testing with the Holm method.

(7) SupFile1 and SupFile2 - which supplementary note or figure are these referring to?

Apologies for this error. These items were meant to link to the FigShare repository where the supplementary files can be found. We have corrected this using the DOI for the repository.

**Reviewer #2 (Recommendations for the authors):**
(1) Authors should clarify that the mouse ENCODE data used for the comparisons was obtained from craniofacial tissue.

This has now been corrected to clarify that the mouse ENCODE data used was from craniofacial tissues. ENCODE mouse embryonic facial prominence ChIP-seq and gene expression quantification file accession numbers and details used in study can be found in Supplementary Table 17.

(2) Given the large differences in TPM for highly expressed genes shown in Figure 5, a MA or volcano plot would provide a more comprehensive view of global transcriptome differences between species.

We have added this plot as Supplementary Figure 13.

(3) It is unclear whether the enrichment analysis was performed for mouse genes, dunnart genes, or both.

In reference to Figure 5, Gene Ontology enrichment analysis was performed on the top 500 highly expressed genes in dunnart. Because there is not an ontology database for dunnart gene IDs, these top 500 dunnart gene IDs were converted to the orthologous gene ID in mouse before performing the enrichment analysis. We apologise for the lack of clarity and have added additional text in the results section to make this clearer. In addition, the relevant methods section now reads:

[New text]: As there is no equivalent gene ontology database for dunnart, we converted the Tasmanian devil RefSeq IDs to Ensembl v103 using biomaRt v2.46.3 and then converted these to mouse Ensembl v103 IDs. In this way we were able to use the mouse Ensembl Gene Ontology annotations for the dunnart gene domains. All gene ontology analyses were performed using clusterProfiler v4.1.4[117], with Gene Ontology from the org.Mm.eg.db v3.12.0 database[118], setting an FDR-corrected p-value threshold of 0.01 for statistical significance.